# Modelling Hierarchical Structure between Dialogue Policy and Natural Language Generator with Option Framework for Task-Oriented Dialogue System

**Jianhong Wang**[*], **Yuan Zhang**[†], **Tae-Kyun Kim**[*♣], **Yunjie Gu**[*♠]

## Abstract

Designing task-oriented dialogue systems is a challenging research topic, since it needs not only to generate utterances fulfilling user requests but also to guarantee the comprehensibility. Many previous works trained end-to-end (E2E) models with supervised learning (SL), however, the bias in annotated system utterances remains as a bottleneck. Reinforcement learning (RL) deals with the problem through using non-differentiable evaluation metrics (e.g., the success rate) as rewards. Nonetheless, existing works with RL showed that the comprehensibility of generated system utterances could be corrupted when improving the performance on fulfilling user requests. In our work, we (1) propose modelling the hierarchical structure between dialogue policy and natural language generator (NLG) with the option framework, called HDNO, where the latent dialogue act is applied to avoid designing specific dialogue act representations; (2) train HDNO via hierarchical reinforcement learning (HRL), as well as suggest the asynchronous updates between dialogue policy and NLG during training to theoretically guarantee their convergence to a local maximizer; and (3) propose using a discriminator modelled with language models as an additional reward to further improve the comprehensibility. We test HDNO on MultiWoz 2.0 and MultiWoz 2.1, the datasets on multi-domain dialogues, in comparison with word-level E2E model trained with RL, LaRL and HDSA, showing improvements on the performance evaluated by automatic evaluation metrics and human evaluation. Finally, we demonstrate the semantic meanings of latent dialogue acts to show the explanability for HDNO.

## 1 Introduction

Designing a task-oriented dialogue system is a popular and challenging research topic in the recent decades. In contrast to the open-domain dialogue system (Ritter et al., 2011), it aims to help people complete real-life tasks through dialogues without human service (e.g., booking tickets) (Young, 2006). In a task-oriented dialogue task, each dialogue is defined with a goal which includes user requests (i.e., represented as a set of key words known as slot values). The conventional task-oriented dialogue system is comprised of 4 modules (see Appendix 3.1), each of which used to be implemented with handcrafted rules (Chen et al., 2017). Given user utterances, it gives responses in turn to fulfill the requests via mentioning corresponding slot values.

Recently, several works focused on training a task-oriented dialogue system in end-to-end fashion (E2E) (Bordes et al., 2016; Wen et al., 2017) for generalizing dialogues outside corpora. To train a E2E model via supervised learning (SL), generated system utterances are forced to fit the oracle responses collected from human-to-human conversations (Budzianowski et al., 2017a). The oracle responses contain faults by humans thus being inaccurate, which leads to biased SL. On the other hand, the goal is absolutely clear, though the criterion of success rate that evaluates the goal completion is non-differentiable and cannot be used as a loss for SL.

---

[*]Imperial College London. [†]Laiye Network Technology Co. Ltd.. [♣]KAIST. [♠]University of Bath. Correspondence to Yunjie Gu: `yg934@bath.ac.uk`.

To tackle this problem, reinforcement learning (RL) is applied to train a task-oriented dialogue system (Williams and Young, 2007; Zhao and Eskénazi, 2016; Peng et al., 2018; Zhao et al., 2019). Specifically, some works merely optimized dialogue policy while other modules, e.g., the natural language generator (NLG), were fixed (Peng et al., 2018; Zhao et al., 2019; Su et al., 2018). In contrast, other works extended the dialogue policy to NLG and applied RL on the entire E2E dialogue system, regarding each generated word in a response as an action (Zhao and Eskénazi, 2016). Although previous works enhanced the performance on fulfilling user requests, the comprehensibility of generated system utterances are corrupted (Peng et al., 2018; Zhao et al., 2019; Tang et al., 2018a). The possible reasons are: (1) solely optimizing dialogue policy could easily cause the biased improvement on fulfilling user requests, ignoring the comprehensibility of generated utterances (see Section 3.1); (2) the state space and action space (represented as a vocabulary) in E2E fashion is so huge that learning to generate comprehensible utterances becomes difficult (Lewis et al., 2017); and (3) dialogue system in E2E fashion may lack explanation during the procedure of decision.

In our work, we propose to model the hierarchical structure between dialogue policy and NLG with the option framework, i.e., a hierarchical reinforcement learning (HRL) framework (Sutton et al., 1999) called HDNO (see Section 4.1) so that the high-level temporal abstraction can provide the ability of explanation during the procedure of decision. Specifically, dialogue policy works as a high-level policy over dialogue acts (i.e. options) and NLG works as a low-level policy over generated words (i.e. primitive actions). Therefore, these two modules are decoupled during optimization with the smaller state space for NLG and the smaller action space for dialogue policy (see Appendix F). To reduce the efforts on designing dialogue act representations, we represent a dialogue act as latent factors. During training, we suggest the asynchronous updates between dialogue policy and NLG to theoretically guarantee their convergence to a local maximizer (see Section 4.2). Finally, we propose using a discriminator modelled with language models (Yang et al., 2018) as an additional reward to further improve the comprehensibility (see Section 5).

We evaluate HDNO on two datasets with dialogues in multiple domains: MultiWOZ 2.0 (Budzianowski et al., 2018) and MultiWOZ 2.1 (Eric et al., 2019), compared with word-level E2E (Budzianowski et al., 2018) trained with RL, LaRL (Zhao et al., 2019) and HDSA (Chen et al., 2019). The experiments show that HDNO works best in the total performance evaluated with automatic metrics (see Section 6.2.1) and the human evaluation (see Section B.1). Furthermore, we study the latent dialogue acts and show the ability of explanation for HDNO (see Section 6.4).

## 2    RELATED WORK

Firstly, we go through the previous works on studying the dialogue act representation for task-oriented dialogue systems. Some previous works optimized dialogue policy with reinforcement learning (RL), which made decision via selecting from handcrafted dialogue acts represented as ontology (Peng et al., 2018; Young et al., 2007; Walker, 2000; He et al., 2018). Such a representation method is easily understood by human beings, while the dialogue act space becomes limited in representation. To deal with this problem, some researchers investigated training dialogue acts via fitting oracle dialogue acts represented in sequence (Chen et al., 2019; Zhang et al., 2019; Lei et al., 2018). This representation method generalized dialogue acts, however, designing a good representation is effort demanding. To handle this problem, learning a latent representation of dialogue act was attempted (Zhao et al., 2019; Yarats and Lewis, 2018). In our work, similar to (Zhao et al., 2019) we learn latent dialogue acts without any labels of dialogue acts. By this view, our work can be regarded as an extension of LaRL (Zhao et al., 2019) on learning strategy.

Then, we review the previous works modelling a dialogue system with a hierarchical structure. In the field of task-oriented dialogue systems, many works lay on modelling dialogue acts or the state space with a hierarchical structure to tackle the decision problem for dialogues with multi-domain tasks (Cuayáhuitl et al., 2009; Peng et al., 2017; Chen et al., 2019; Tang et al., 2018b; Budzianowski et al., 2017b). Distinguished from these works, our work views the relationship between dialogue policy and natural language generator (NLG) as a natural hierarchical structure and models it with the option framework (Sutton et al., 1999). In the field of open-domain dialogue system, a similar hierarchical structure was proposed (Serban et al., 2017; Saleh et al., 2019) but with a different motivation from ours. In this sense, these two fields are possible to be unified.

Finally, among the works training with hierarchical reinforcement learning (HRL), some of them set up an extrinsic reward for high-level policy and an intrinsic reward for low-level policy respectively to encourage the convergence (Peng et al., 2017; Budzianowski et al., 2017b). In our work, we train both high-level policy and low-level policy with identical rewards to guarantee the consistency between two policies (Sutton et al., 1999). On the other hand, in the field of open-domain dialogue system, Saleh et al. (2019) represented the joint generated utterances over a turn as a low-level action such that both high-level policy and low-level policy were in identical time scales. Besides, its low-level policy gradients flew through high-level policy during training, which degraded hierarchical policies to an E2E policy with a word-level action space. In our work, (1) dialogue policy and NLG are decoupled during optimization and no gradients are allowed to flow between them; (2) these two policies are asynchronously updated to theoretically guarantee the convergence to a local maximizer; and (3) each generated word is regarded as a low-level action.

## 3 BACKGROUND

### 3.1 TASK-ORIENTED DIALOGUE SYSTEM

**Brief Introduction:** A task-oriented dialogue system aims to help fulfill a user's task through conversation in turns. In general, each dialogue is modelled with an ontology called goal which includes inform slots and request slots. The traditional modular dialogue system is constituted of natural language understanding (NLU), dialogue state tracker (DST), dialogue policy and natural language generator (NLG). For a dialogue system, it needs to infer inform slots from user utterances and transform them to a dialogue state, which is completed by NLU and DST (Chen et al., 2017). In this work, we focus on optimizing dialogue policy and NLG, leveraging oracle dialogue states and database search results to produce dialogue acts and then responses (that should include as many request slots as possible) in turns. For optimizing dialogue policy, it is modelled with Markov decision process (MDP) (Williams and Young, 2007).

**Existing Challenges:** We identify the main challenges of task-oriented dialogue systems: (1) A dialogue with a single domain (i.e. completing one task in a dialogue) has been broadly studied, however, handling a dialogue with multiple domains is more challenging and needs more studies on it (Budzianowski et al., 2018); (2) If ignoring the syntactic structure of generated system utterances (i.e. losing comprehensibility), the mission of task-oriented dialogues will be simplified to generating corresponding labels (i.e., slots) for user utterances. Several existing algorithms already reached high scores on request slots acquisition but low scores on the comprehensibility of generated system utterances (Zhao et al., 2019; Mehri et al., 2019), so the simplified task has been well-addressed. Reversely, if only focusing on the comprehensibility, the score on request slots acquisition could be drastically affected (Chen et al., 2019; Hosseini-Asl et al., 2020). In this work, we investigate the trade-off between the comprehensibility and request slots acquisition; (3) Designing and annotating a dialogue act structure is effort demanding (Budzianowski et al., 2018). Therefore, learning a meaningful latent dialogue act becomes a new challenge (Zhao et al., 2019).

### 3.2 HIERARCHICAL REINFORCEMENT LEARNING WITH OPTION FRAMEWORK

Hierarchical reinforcement learning (HRL) is a variant of reinforcement learning (RL) which extends the decision problem to coarser grains with multiple hierarchies (Sutton et al., 1999; Dayan and Hinton, 1993; Parr and Russell, 1998; Dietterich, 1998). Amongst several HRL methods, the option framework (Sutton et al., 1999) is a temporal abstraction for RL, where each option (i.e. a high-level action) lasts for a number of steps through primitive actions (i.e. low-level actions). From the view of decision problems, an MDP defined with a fixed set of options naturally forms a semi-MDP (SMDP). Formally, an option $\boldsymbol{o} = \langle \mathbb{I}, \beta, \pi \rangle$ is composed of three components: an initiation set $\mathbb{I} \subseteq \mathbb{S}$ (where $\mathbb{S}$ is a state space), a termination function $\beta(\boldsymbol{s}_t) \mapsto [0,1]$ and an intra-option policy $\pi(\boldsymbol{a}_t|\boldsymbol{s}_t) \mapsto [0,1]$. The reward over primitive actions is defined as $r_t \in \mathbb{R}$, identical to vanilla RL. An option $\boldsymbol{o}_t$ is available at $\boldsymbol{s}_t \in \mathbb{I}$. At each $\boldsymbol{s}_t$, $\pi$ is used to decide a low-level action $\boldsymbol{a}_t$ until the option is stochastically terminated by $\beta$. Similar to RL on flat actions, the probability transition function over options is defined as $p(\boldsymbol{s}'|\boldsymbol{s}_t, \boldsymbol{o}_t) = \sum_{k=1}^{\infty} p(\boldsymbol{s}', k) \, \gamma^k$, where $p(\boldsymbol{s}', k)$ is the probability of an option terminating in k steps and $\gamma \in (0, 1)$. The policy over options is defined as $\mu(\boldsymbol{o}_t|\boldsymbol{s}_t) \mapsto [0, 1]$ and the reward over options lasting for $k$ steps is defined as $g(\boldsymbol{s}_t, \boldsymbol{o}_t, \boldsymbol{s}_{t+k}) = \mathbb{E}_\pi \big[ \sum_{i=t}^{t+k} \gamma^{i-t} \, r_i \big]$

(abbreviated as $g_t$ for simplicity). Given a set of options $o \in \mathbb{O}$, the optimization problem over options is defined as $\max_o \mathbb{E}_o[ \sum_{k \in M} \gamma^{k-t} g_k ]$, where $\boldsymbol{m} = (t, t', ...)$ is a sequence containing the time step of each event [1] that will be experienced from some time step $t$ to future. To automatically fit complicated circumstances, we may also need to discover options dynamically during learning. Intra-option policy gradient theorem and termination policy gradient theorem (Bacon et al., 2017) provide the basis to apply a policy gradient method for option discovery.

# 4 MODELLING HIERARCHICAL STRUCTURE BETWEEN DIALOGUE POLICY AND NLG WITH OPTION FRAMEWORK

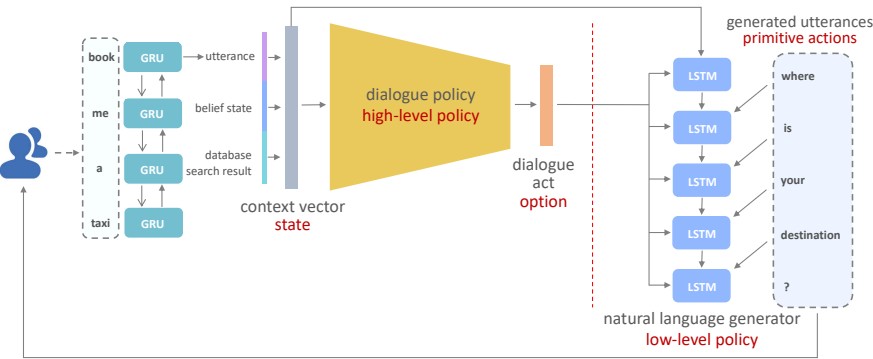

Figure 1: This diagram demonstrates the overall architecture for modelling the hierarchical structure between dialogue policy (i.e. high-level policy) and NLG (i.e. low-level policy) as an option framework. The text in gray represents the concepts for a traditional task-oriented dialogue system whereas the text in red matches the concepts for the option framework.

## 4.1 MODEL FORMULATION

In this section, we present a view on modelling the **H**ierarchical structure between **D**ialogue policy and **N**atural language generator (NLG) with the **O**ption framework (Sutton et al., 1999), called HDNO. Specifically, a dialogue act in HDNO is seen as an option whereas each generated word from NLG is a primitive action. Accordingly, dialogue policy and NLG become the policy over option (i.e. high-level policy) and the intra-option policy (i.e. low-level policy) respectively. Distinguished from a conventional modular system, we additionally give a context to NLG to satisfy the conditions of the option framework. Moreover, since the primitive action space (i.e. a vocabulary) comprises a termination symbol, NLG can take over the responsibility of termination. For this reason, termination policy in the original option framework is absorbed into the intra-option policy. The formal definition of HDNO is shown in Definition 1.

**Definition 1.** *A dialogue policy (i.e. a policy over option) is defined as* $\phi : \mathbb{S} \times \mathbb{O} \rightarrow [0, 1]$*, where* $\mathbb{S}$ *is a set of contexts (i.e. utterances, dialogue states and database search results);* $\mathbb{O}$ *is a set of dialogue acts (i.e. options). A dialogue act is defined as* $\boldsymbol{o} = \langle \mathbb{I}_{\boldsymbol{o}}, \pi_{\boldsymbol{o}} \rangle$*, where* $\mathbb{I}_{\boldsymbol{o}} \subseteq \mathbb{S}$ *is a set of corresponding contexts for a generated word and* $\pi_{\boldsymbol{o}} : \mathbb{I}_{\boldsymbol{o}} \times \mathbb{V} \rightarrow [0, 1]$ *is natural language generator (NLG) (i.e. an intra-option policy).* $\mathbb{V}$ *is a vocabulary (including a termination symbol).*

According to MDP theorem over option (Sutton et al., 1999) and intra-option policy gradient theorem (Bacon et al., 2017), we can naturally apply REINFORCE (Williams, 1992) to learn both $\phi$ and $\pi$. Therefore, following Section 3 and Definition 1, we can write policy gradients in our case such that $\nabla J(\phi) = \mathbb{E}_\phi[ \sum_{k \in M} \gamma^{k-t} g_k \nabla \ln \phi(\boldsymbol{o}_t | \boldsymbol{s}_t) ]$ and $\nabla J(\pi_{\boldsymbol{o}_t}) = \mathbb{E}_{\pi_{\boldsymbol{o}_t}}[ \sum_{i=t}^{T} \gamma^{i-t} r_i \nabla \ln \pi_{\boldsymbol{o}_t}(\boldsymbol{w}_t | \boldsymbol{s}_t) ]$, where we assume that the length of all generated system utterances is $T$ and $\boldsymbol{m} = (t, t', ...)$ is a sequence containing the time steps of an event that appear in future for an arbitrary $\boldsymbol{o}_t = \langle \mathbb{I}_{\boldsymbol{o}_t}, \pi_{\boldsymbol{o}_t} \rangle \in \mathbb{O}$.

---

[1] An event is defined as a policy over option calling an intra-option policy at some state.

## 4.2 Asynchronously Updating Dialogue Policy and NLG during Learning

As mentioned in Section 4.1, dialogue policy and NLG are written as $\phi(\boldsymbol{o}|\boldsymbol{s})$ and $\pi_{\boldsymbol{o}}(\boldsymbol{w}|\boldsymbol{s})$ respectively. However, since $\boldsymbol{o} = \langle \mathbb{I}_{\boldsymbol{o}}, \pi_{\boldsymbol{o}} \rangle$, we can assume that when dialogue policy made a decision, it has to consider the current performance on the overall set of low-level policies for NLG, denoted as $\pi = \{\pi_{\boldsymbol{o}}\}_{\boldsymbol{o} \in \mathbb{O}}$. For the reason, we temporarily rewrite dialogue policy to $\phi(\boldsymbol{o}|\pi, \boldsymbol{s})$ for convenience. The aim is finding the best policies (i.e. maximizers) so that the value can be maximized such that $\max_{\phi, \pi} v(\boldsymbol{s}|\phi(\boldsymbol{o}|\pi, \boldsymbol{s})), \forall \boldsymbol{s} \in \mathbb{S}$. If updating these two policies synchronously, it will cause the composite state (i.e. $\langle \pi, \boldsymbol{s} \rangle$) of $\phi(\boldsymbol{o}|\pi, \boldsymbol{s})$ inconsistent before and after the update each time. Therefore, the value does not always monotonically improve during learning, which will affect the convergence of both policies (see Proposition 1). To address this problem, we suggest updating dialogue policy and NLG asynchronously during learning to theoretically guarantee the convergence of these policies to a local maximizer (see Proposition 2). The proofs of these two propositions are left to appendices due to limited space.

**Proposition 1.** *Following the model of Definition 1, if $\phi(\boldsymbol{o}|\pi, \boldsymbol{s})$ and $\pi$ are synchronously updated, the value does not always monotonically improve and the policies may never converge to a local maximizer.*

**Assumption 1.** *(1) Reward function is bounded. (2) With sufficient number of samples, the Monte Carlo estimation for value on any state is accurate enough.*

**Proposition 2.** *Following the model of Definition 1 and Assumption 1, if $\phi(\boldsymbol{o}|\pi, \boldsymbol{s})$ and $\pi$ are asynchronously updated, the value can improve monotonically during learning and the policies can finally converge to a local maximizer.*

## 4.3 Implementation of HDNO

In implementation of HDNO, we represent a dialogue act as latent factors (Zhao et al., 2019), which reduces the effort on designing a suitable representation. In detail, a dialogue act $\boldsymbol{z}$ (i.e. an indicator representing an option) is sampled from a dialogue policy represented as an isotropic multivariate Gaussian distribution such that $\phi(\boldsymbol{z}|\boldsymbol{c}; \lambda) = \mathcal{N}(\boldsymbol{z}|\boldsymbol{\mu}(c), \boldsymbol{\Sigma}(c))$, where $\boldsymbol{c}$ is a context as well as $\boldsymbol{\mu}(\boldsymbol{c}) \in \mathbb{R}^{K}$ and $\boldsymbol{\Sigma}(\boldsymbol{c}) \in \mathbb{R}^{K \times K}$ are parameterized with $\lambda$. Moreover, NLG, i.e. $\pi(\boldsymbol{w}_t|\boldsymbol{z}, \tilde{\boldsymbol{c}}_t; \nu)$, is represented as a categorical distribution over a word parameterized with $\nu$, conditioned on an option $\boldsymbol{z}$ and a context $\tilde{\boldsymbol{c}}_t$ which involves preceding generated utterances in addition to the context $\boldsymbol{c}$ that activates the option $\boldsymbol{z}$. The full picture of this architecture is shown in Figure 1.

Furthermore, $\phi(\boldsymbol{z}|\boldsymbol{c}; \lambda)$ (i.e. dialogue policy) is implemented with one-layer linear model and outputs the mean and variance of a multivariate Gaussian distribution. The input of $\phi(\boldsymbol{z}|\boldsymbol{c}; \lambda)$ is a context vector. In details, the last user utterances are firstly encoded with a bidirectional RNN (Schuster and Paliwal, 1997) with gated recurrent unit (GRU) cell (Chung et al., 2014) and global type attention mechanism (Bahdanau et al., 2015). Then, an oracle dialogue state and an oracle database search result are concatenated to an encoding vector of user utterances to form a context vector. The utterance encoder is only trained during pretraining and fixed as a context extractor during HRL, so that the context space is reduced. On the other hand, $\pi(\boldsymbol{w}_t|\boldsymbol{z}, \tilde{\boldsymbol{c}}_t; \nu)$ (i.e. NLG) is implemented with a recurrent neural network (RNN) with long short-term memory (LSTM) cell (Hochreiter and Schmidhuber, 1997), where the initial state is the concatenation of a context vector and a dialogue act sampled from dialogue policy. The information in the initial state is assumed to be propagated to the hidden states at the future time steps, so we only feed it in the initial state in implementation.

## 4.4 Pretraining with Bayesian Framework

Compared to VHRED (Serban et al., 2017) proposed in the field of open-domain dialogue systems, the latent dialogue act in HDNO is equivalent to the latent variables in VHRED. However, a context of HDNO includes not only user utterances but also dialogue state and database search result. In this sense, HDNO extends VHRED to the field of task-oriented dialogue systems. As a result, by changing user utterances in VHRED to the context in HDNO, we can directly formulate a variational lower bound following the Bayesian framework and the model in Section 4.1 such that

$$\max_{\lambda, \nu} \mathbb{E}_{\boldsymbol{z} \sim \phi(\boldsymbol{z}|\boldsymbol{c}; \lambda)} \Big[ \sum_{t=1}^{T} \log \pi(\boldsymbol{w}_t|\boldsymbol{z}, \tilde{\boldsymbol{c}}_t; \nu) \Big] - \beta \, \mathrm{KL}[\, \phi(\boldsymbol{z}|\boldsymbol{c}; \lambda) \,||\, \mathcal{N}(\boldsymbol{z}|\boldsymbol{0}, \boldsymbol{I}) \,], \qquad (1)$$

where $\phi(\boldsymbol{z}|\boldsymbol{c};\lambda)$ is constrained by KL-divergence (Kullback and Leibler, 1951) from a multivariate standard Gaussian distribution $\mathcal{N}(\boldsymbol{z}|\boldsymbol{0}, \boldsymbol{I})$. Referring to (Higgins et al., 2017), we additionally add a multiplier $\beta$ on the term of KL-divergence to control the disentanglement of a latent dialogue act $\boldsymbol{z}$. We use Eq.1 for pretraining dialogue policy and NLG in E2E fashion with oracle system utterances to roughly allocate the roles of these two modules. Nevertheless, restricted with the existing human faults in oracle system utterances (see Section 1), we need to further improve it via HRL.

## 5  USING DISCRIMINATOR OF LANGUAGE MODELS AS A REWARD

Benefiting from the availability of non-differentiable evaluation metrics in RL, we can directly apply the success rate on reward denoted by $r^{\text{succ}}$. However, there exists two potential issues: (1) since the success rate is given until the end of a dialogue that is zero in the other turns, which may cause a sparse reward; and (2) the success rate is only correlated to the occurrence of request slots in generated system utterances, the improvement on comprehensibility may be weakened. To mitigate the above drawbacks, we propose to leverage a discriminator modelled as language models (see Definition 2) as an additional reward. Specifically, at each time step it evaluates each generated word by log-likelihood to reflect the comprehensibility.

**Definition 2.** *Discriminator $D(\boldsymbol{w}_t|\boldsymbol{w}_{t-1}) \mapsto [0,1]$ is defined as the Markov language model following (Yang et al., 2018). At some time step $\tau$, for an arbitrary option $\boldsymbol{e} = \langle \mathbb{I}_e, \pi_e \rangle$, $\boldsymbol{w}_\tau \sim \pi_e(\cdot|\boldsymbol{s}_\tau)$ is a sampled word at state $\boldsymbol{s}_\tau \in \mathbb{I}_e$. The reward of discriminator to evaluate $\boldsymbol{w}_\tau$ is defined as $r_\tau^{disc} = \log D(\boldsymbol{w}_\tau|\boldsymbol{w}_{\tau-1})$, where $\boldsymbol{w}_{\tau-1}$ is a generated word at time step $\tau - 1$.*

According to Definition 2, we can see that $\sum_{t=1}^{T'} \gamma^t r^{\text{disc}}$ consistently grows as the joint log-likelihood $\sum_{t=1}^{T'} \log D(\boldsymbol{w}_t|\boldsymbol{w}_{t-1})$ grows, thereby maximizing $\mathbb{E}_\pi[\sum_{t=1}^{T'} \gamma^t r_t^{\text{disc}}]$ is almost equivalent to maximizing $\mathbb{E}[\sum_{t=1}^{T'} \log D(\boldsymbol{w}_t|\boldsymbol{w}_{t-1})]$ when $\gamma$ is around 1 and $T'$ is not too large, where $T'$ denotes the number of time steps in a turn. For this reason, $r^{\text{disc}}$ is suitable for evaluating the comprehensibility of generated system utterances if we presume that the discriminator can well represent human language. Combining the reward of success rate $r^{\text{succ}}$ and the reward of discriminator $r^{\text{disc}}$, we propose a total reward such that

$$r_t^{\text{total}} = (1 - \alpha)\, r_t^{\text{succ}} + \alpha\, r_t^{\text{disc}}, \tag{2}$$

where $\alpha \in [0, 1]$ is a multiplier controlling the trade-off between these two types of rewards. In implementation, the discriminator is equipped with the same architecture as that of NLG.

## 6  EXPERIMENTS

### 6.1  EXPERIMENTAL SETUPS

**Dataset Description:** To evaluate the performance of our task-oriented dialogue system, we run experiments on the latest benchmark datasets MultiWoz 2.0 (Budzianowski et al., 2018) and Multi-Woz 2.1 (Eric et al., 2019). MultiWoz 2.0 is a large scale task-oriented dialogue dataset including 10425 dialogues that spans 7 distinct domains, where the whole dialogues are generated by human-to-human conversations. Each dialogue is defined with a goal for a user, which may be consisting of 1-5 domains. A dialogue system attempts to fulfill a goal by interacting with a user. As for data preprocessing, we follow the same delexicalized method provided by (Budzianowski et al., 2018), also used in other works (Zhao et al., 2019; Chen et al., 2019). On the other hand, MultiWoz 2.1 is a modified version of MultiWoz 2.0 which mainly fixes the noisy dialogue state annotations and corrects 146 dialogue utterances. Finally, the dataset of either MultiWoz 2.0 or MultiWoz 2.1 is split into a training set with 8438 dialogues, a validation set with 1000 dialogues and a testing set with 1000 dialogues (Budzianowski et al., 2018; Eric et al., 2019).

**Task Description:** Since we only concentrate on learning dialogue policy and natural language generator (NLG), all experiments are conducted on the dialog-context-to-text generation task proposed in (Budzianowski et al., 2018). In this task, it assumes that a dialogue system has access to the oracle dialogue state and database search result. Given user utterances, a dialogue system attempts to generate appropriate utterances as a response in each turn. To train a dialogue system with hierarchical

reinforcement learning (HRL), we follow the setups described in Definition 1. Each dialogue is only evaluated with the goal (e.g. calculating the success rate) at the end of dialogue, which means that no evaluation is allowed during any of turns.

**Automatic Evaluation Metrics:** Following (Budzianowski et al., 2017a), we leverage three automatic metrics to evaluate generated utterances from a dialogue system such as inform rate, success rate and BLEU score. Inform rate measures whether a dialogue system provides appropriate entities (e.g., the name of restaurant). Success rate shows the ratio of request slots appearing in generated utterances. BLEU score (Papineni et al., 2002) evaluates the comprehensibility of generated utterances. Finally, we use a popular total score (Zhang et al., 2019; Mehri et al., 2019) such that $0.5 \times$ (Inform + Success) + BLEU to fairly evaluate the performance of a dialogue system.

**Baseline Description:** We compare HDNO with other models such as LaRL (Zhao et al., 2019), HDSA (Chen et al., 2019), and a baseline end-to-end model (Budzianowski et al., 2018). All these models leveraged oracle dialogue states and database search results, which are introduced as follows:

- The baseline end-to-end model (Budzianowski et al., 2018) is a model which directly maps a context to system utterances. Followed by (Zhao et al., 2019), we train it with RL, where each generated word is looked as an action. For convenience, we name it as word-level E2E model, abbreviated as WE2E.

- LaRL (Zhao et al., 2019) is a model that firstly represents a dialogue act as latent factors in the field of task-oriented dialogue system. Specifically, it models a latent dialogue act as categorical variables, each of which is mapped to a continuous embedding vector for learning. During training, it only updates dialogue policy, where a latent categorical dialogue act for each turn is looked as an action.

- HDSA (Chen et al., 2019) is a model that represents each dialogue act as a hierarchical graph. To fit the oracle dialogue act, a pretrained 12-layer BERT (Devlin et al., 2019) is applied. Then the predicted dialogue act is transformed to the hierarchical graph structure with 3-layer self-attention model (Vaswani et al., 2017), called disentangled self-attention model. This model is trained only with supervised learning (SL).

**Experimental Details:** For HDNO [2], we pretrain a model following Eq.1 and select the best model with the minimum loss on the validation set; as well as the discriminator is pretrained with oracle system utterances. During HRL, we initialize parameters with the pretrained model and select the best model according to the greatest reward on the validation set. For efficiency, we only use greedy search for decoding in validation. In test, we apply beam search (Medress et al., 1977) for decoding to obtain a better performance. The beam width is selected through the best validation performance for each model. For simplicity, we only show the performance of the best beam width on test set. We use stochastic gradient descent (SGD) for HRL and Adam optimizer (Kingma and Ba, 2015) for pretraining. For the baselines, we train them with the original source codes. The specific details and hyperparameters for training and testing are shown in Appendix B.3.

**Notification:** Please notice that the results of baselines showed in their original papers could be underestimated, which is due to the upgrade of the official evaluator this year [3]. For this reason, we re-run these experiments with the original open source codes and evaluate the performance of all models (including HDNO) via the latest official evaluator in this work.

## 6.2 MAIN RESULTS

### 6.2.1 AUTOMATIC EVALUATION METRICS

We firstly compare HDNO with the state-of-the-art baselines and the human performance on both datasets via automatic evaluation metrics. As Table 1 shows, we can see that HDNO trained with the proposed asynchronous updates between dialogue policy and NLG (i.e. HDNO (Async.)) largely outperforms the baselines on the inform rate and total score, while its BLEU score is lower than that of HDSA. Moreover, the performance of all models trained with RL except for HDNO trained

---

[2]The source code of implementation of HDNO is on `https://github.com/mikezhang95/HDNO`.

[3]Please check it via the clarification for this incident below the table called Policy Optimization on the official website of MultiWoz: `https://github.com/budzianowski/multiwoz`.

Table 1: The table shows the main results on MultiWoz 2.0 and MultiWoz 2.1 evaluated with the automatic evaluation metrics. The results of HDNO are from the models trained with the proposed reward shape in Section 5, where $\alpha = 0.0001$ for MultiWoz 2.0 and $\alpha = 0.01$ for MultiWoz 2.1.

| | MultiWoz 2.0 | | | | MultiWoz 2.1 | | | |
|---|---|---|---|---|---|---|---|---|
| | Inform(%) | Success(%) | BLEU(%) | Total | Inform(%) | Success(%) | BLEU(%) | Total |
| Human | 91.00 | 82.70 | - | - | 86.30 | 79.10 | - | - |
| WE2E | 90.29 | **86.59** | 14.08 | 102.52 | 90.89 | 83.58 | 14.52 | 101.76 |
| LaRL | 93.49 | 84.98 | 12.01 | 101.25 | 92.39 | **85.29** | 13.72 | 102.56 |
| HDSA | 88.90 | 73.40 | **23.15** | 104.30 | 85.60 | 75.50 | **21.57** | 102.12 |
| Pretraining (Bayesian) | 69.50 | 62.00 | 19.10 | 84.85 | 71.40 | 62.80 | 19.12 | 86.22 |
| HDNO (Sync.) | 83.20 | 73.50 | 19.82 | 98.17 | 83.10 | 70.80 | 18.81 | 95.76 |
| HDNO (Async.) | **96.40** | 84.70 | 18.85 | **109.40** | **92.80** | 83.00 | 18.97 | **106.77** |

with synchronous updates between dialogue policy and NLG (i.e. HDNO (Sync.)) on the inform rate and success rate exceeds that of the model trained with SL (i.e. HDSA). The possible reason may be that SL is highly dependent on the oracle system utterances and humans may commit faults during generating these dialogues as we stated in Section 1. Besides, the poor results on HDNO (Sync.) validates the theoretical analysis in Section 4.2 that synchronous updates between dialogue policy and NLG could cause the failure to approach a local optimum, while HDNO (Async.) shows the success for the asynchronous updates proposed in Proposition 2. Furthermore, the results of pretraining give the evidence that the improvement is actually from the proposed algorithm. For conciseness, we write HDNO instead of HDNO (Async.) in the rest of paper. We also conduct human evaluations for WE2E, LaRL and HDNO that are shown in Appendix B.1.

## 6.3 STUDY ON REWARD SHAPES

Table 2: The table shows the results of different reward shapes for HDNO on MultiWoz 2.0.

| | $\alpha$ | Inform(%) | Success(%) | BLEU(%) | Total |
|---|---|---|---|---|---|
| Success | - | 96.10 | 84.20 | 18.51 | 108.66 |
| Success + BLEU | - | 95.60 | 83.30 | 18.99 | 108.44 |
| | 0.0001 | 96.40 | 84.70 | 18.85 | **109.40** |
| Success + Discriminator | 0.0005 | 96.30 | **84.90** | 18.50 | 109.10 |
| | 0.001 | 96.20 | 84.20 | **19.04** | 109.24 |
| | 0.005 | **97.00** | 84.10 | 18.72 | 109.27 |

We now compare the proposed reward shape in Eq.2 with a reward only constituted of the success rate and a reward combining the success rate with BLEU score (i.e. a linear combination similar to the proposed reward shape). As Table 2 shows, in comparison with other reward shapes, the proposed reward shape (i.e. success + discriminator) performs better on preserving the comprehensibility while improving the success rate and inform rate to maximum. To further study the impact of discriminator in the total reward, we also run several ablation studies on $\alpha$ (see Section 5). As Table 2 shows, the results oscillate within a small range, which means that the proposed reward shape is not sensitive to the hyperparameter $\alpha$ if it is selected within a rational range.

## 6.4 STUDY ON LATENT DIALOGUE ACT

We now study the semantic meanings of latent dialogue acts and demonstrate the clustering results as Figure 2 shows. To show the explanability of HDNO that other baselines do not possess, we show the results for both HDNO and LaRL. Since LaRL tends to generate duplicate latent dialogue acts, the dots in the diagram are overlapped. Through analyzing the randomly selected system utterances, we find that the clusters of latent dialogue acts of HDNO possesses some semantic meanings, while that of LaRL is difficult to be observed any meaningful explanations. Next, we briefly describe our findings on clusters of HDNO. The clusters in blue dots and green dots are related to the general phrases for goodbye at the end of service; the cluster in orange dots is related to the booking for trains; the cluster in red dots is related to informing user with database search results; the cluster in brown dots is related to recommendation; the cluster in pink dots is related to informing unsuccessful booking; the cluster in grey is related to informing booked; and the cluster in yellow is related

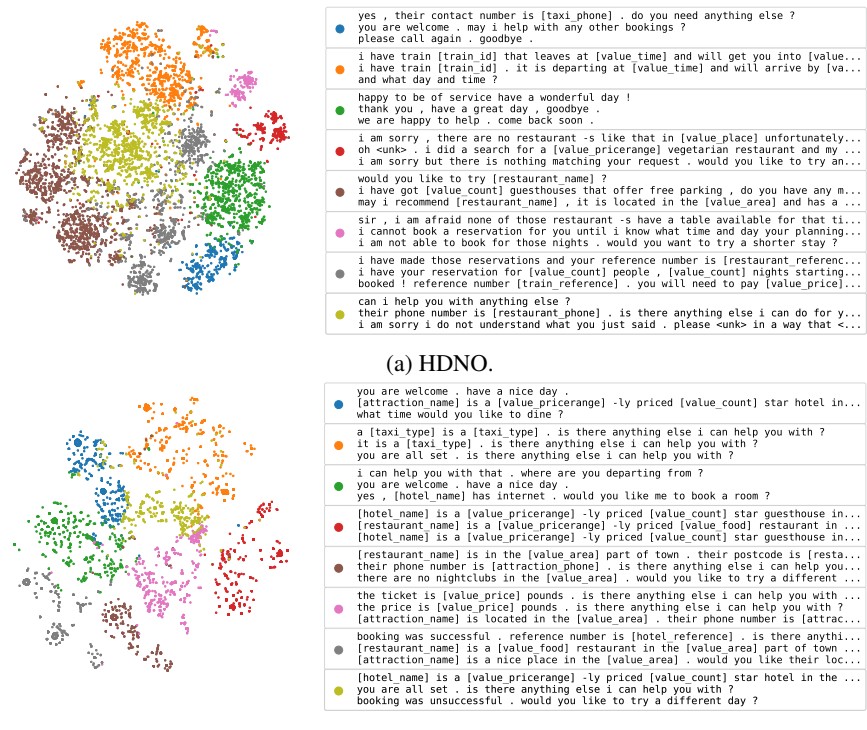

(a) HDNO.

(b) LaRL.

Figure 2: These diagrams demonstrate latent dialogue acts of HDNO and LaRL clustered in 8 categories on a 2-D plane. Clustering is conducted with k-means algorithm (Arthur and Vassilvitskii, 2006) on the original dimensions whereas dimension reduction is conducted with T-SNE algorithm (Maaten and Hinton, 2008). We randomly show 3 turns of system utterances for each cluster.

to requesting more information. Surprisingly, these semantic meanings of latent dialogue acts are highly correlated with that of the oracle handcrafted dialogue acts (see Appendix E) described by Budzianowski et al. (2018). Therefore, learning latent dialogue act with the option framework (Sutton et al., 1999) may potentially substitute for handcrafting dialogue act with ontology, without losing its explanability.

# 7    CONCLUSION AND FUTURE WORK

In this paper, we present a view on modelling the hierarchical structure between dialogue policy and natural language generator (NLG) with the option framework (Sutton et al., 1999) in a task-oriented dialogue system and train it with hierarchical reinforcement learning (HRL). Moreover, we suggest asynchronous updates between dialogue policy and NLG to theoretically guarantee their convergence to a local maximizer. Finally, we propose using a discriminator modelled as language models (Yang et al., 2018) as a reward to further improve the comprehensibility of generated responses.

In the future work, we are going to extend this work to optimizing all modules by HRL instead of only dialogue policy and NLG, as well as study on solving the problem of credit assignment among these modules (Chen et al., 2017) during training. Moreover, thanks to the option framework (Sutton et al., 1999), the latent dialogue act shows explicit semantic meanings, while disentangling factors of a latent dialogue act (i.e. each latent factor owning a distinct semantic meaning) during HRL is left to be further investigated.

## ACKNOWLEDGMENTS

This work is supported by the Engineering and Physical Sciences Research Council of UK (EPSRC) under awards EP/S000909/1.

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

## APPENDICES

## A  PROOFS

### A.1  PROOF OF PROPOSITION 1

**Proposition 1.** *Following the model of Definition 1, if $\phi(\boldsymbol{o}|\pi, \boldsymbol{s})$ and $\pi$ are synchronously updated, the value does not always monotonically improve and the policies may never converge to a local maximizer.*

*Proof.* At an arbitrary time step $t \in \mathbb{N}$, assume that both of $\phi(\boldsymbol{o}|\pi, \boldsymbol{s})$ and $\pi$ have been updated for $q \in \mathbb{N}$ times, denoted as $\phi^q(\pi^q)$ and $\pi^q$ respectively for conciseness. The current value for any arbitrary state $\boldsymbol{s} \in \mathbb{S}$ is denoted as $v(\boldsymbol{s}|\phi^q(\pi^q))$. If we synchronously update $\phi^q(\pi^q)$ and $\pi^q$, we can obtain the following equations on value after updates at time step $t \in \mathbb{N}$ such that

$$\begin{cases} v^t\big(\boldsymbol{s}|\phi^{q+1}(\pi^q)\big) \geq v^t\big(\boldsymbol{s}|\phi^q(\pi^q)\big), \\ v^t\big(\boldsymbol{s}|\phi^q(\pi^{q+1})\big) \geq v^t\big(\boldsymbol{s}|\phi^q(\pi^q)\big). \end{cases} \tag{3}$$

At next time step $t + 1$, however, the actual value for any arbitrary state $s$ that we obtain from the last synchronous update is the equation such that

$$v^{t+1}\big(s|\phi^{q+1}(\pi^{q+1})\big), \tag{4}$$

and the following scenario such that

$$v^{t+1}\big(s|\phi^{q+1}(\pi^{q+1})\big) < v^{t+1}\big(s|\phi^{q}(\pi^{q})\big) \tag{5}$$

could happen, which means that the monotonic improvement path of the value cannot always hold during learning. Since the value is an evaluation of policies, the policies could never converge to a local maximizer. $\square$

## A.2 PROOF OF PROPOSITION 2

**Lemma 1** (Bauschke et al. (2011)). *(1) Any increasing bounded sequence $(x_n)_{n \in \mathbb{N}}$ in $\mathbb{R}$ is Fejér Monotone with respect to $\sup\{x_n\}_{n \in \mathbb{N}}$ such that $\forall n \in \mathbb{N}$, $||x_{n+1} - \sup\{x_n\}_{n \in \mathbb{N}}|| \leq ||x_n - \sup\{x_n\}_{n \in \mathbb{N}}||$. (2) For a Fejér Monotone Sequence $(x_n)_{n \in \mathbb{N}}$, $||x_n - \sup\{x_n\}_{n \in \mathbb{N}}||_{n \in \mathbb{N}}$ converges, $|| \cdot ||$ is an arbitrary norm in $\mathbb{R}$.*

**Assumption 1.** *(1) Reward function is bounded. (2) With sufficient number of samples, the Monte Carlo estimation for value on any state is accurate enough.*

**Proposition 2.** *Following the model of Definition 1 and Assumption 1, if $\phi(o|\pi, s)$ and $\pi$ are asynchronously updated, the value can improve monotonically during learning and both policies can finally converge to a local maximizer.*

*Proof.* Firstly, in REINFORCE (Williams, 1992) the value is approximated by Monte Carlo estimation. Following (2) in Assumption 1 and the denotation shown in the proof of Proposition 1 above, if we update $\phi(\pi)$ and $\pi$ asynchronously every $k \in \mathbb{N}$ steps, from some time step $t \in \mathbb{N}$, assuming that both of policies have been updated $q \in \mathbb{N}$ times, we can construct a monotonically increasing sequence of values for any arbitrary state $s \in \mathbb{S}$ during learning such that

$$v^t\big(s|\phi^q(\pi^q)\big) \leq v^{t+k}\big(s|\phi^{q+1}(\pi^q)\big) \leq v^{t+2k}\big(s|\phi^{q+1}(\pi^{q+1})\big)$$

$$\leq v^{t+3k}\big(s|\phi^{q+2}(\pi^{q+1})\big) \leq \cdots \leq \begin{cases} v^{t+nk}\big(s|\phi^{q+\frac{n}{2}}(\pi^{q+\frac{n}{2}})\big), & \text{if n is an even number,} \\ v^{t+nk}\big(s|\phi^{q+\frac{n+1}{2}}(\pi^{q+\frac{n-1}{2}})\big), & \text{if n is an odd number.} \end{cases} \tag{6}$$

Due to (1) in Assumption 1, we get that the value is bounded and we consider that $v(s|\phi(\pi)) \in \mathbb{R}$, $\forall s \in \mathbb{S}$. According to (1) Lemma 1, the sequence of values is Fejér Monotone with respect to the maximum value. For simplicity, we denote the sequence of values in Eq.6 as $\{v^m\}_{m \in \{t+nk|t,b,k \in \mathbb{N}\}}$ and the maximum value as $v^*$. Since (2) in Lemma 1, we can conclude that $\{||v^m - v^*||\}_{m \in \{t+nk|t,b,k \in \mathbb{N}\}}$ can converge. Also, since the $v^*$ has to be the final item of the sequence, we can write that

$$\big| \big| ||v^m - v^*|| - ||v^* - v^*|| \big| \big| \to 0, \ m \to \infty. \tag{7}$$

If we rearrange the left hand side of Eq.7, we can obtain the result such that

$$v^m \to v^*, \ m \to \infty. \tag{8}$$

From Eq.8, we can conclude that finally the sequence of $v(s|\phi(\pi))$ will converge to some local maximum. Since the value is an evaluation of $\phi(o|\pi, s)$ and $\pi(w|s)$, we can get the conclusion that the asynchronous updates enable these two policies to converge to a local maximizer. $\square$

# B EXTRA EXPERIMENTAL RESULTS

## B.1 HUMAN EVALUATION

Due to the possible inconsistency between automatic evaluation metrics and human perception, we conducted human evaluation on comparing the quality of generated responses. We provide two criteria for human assessors to evaluate the generated responses: (1) fluency: how fluent the

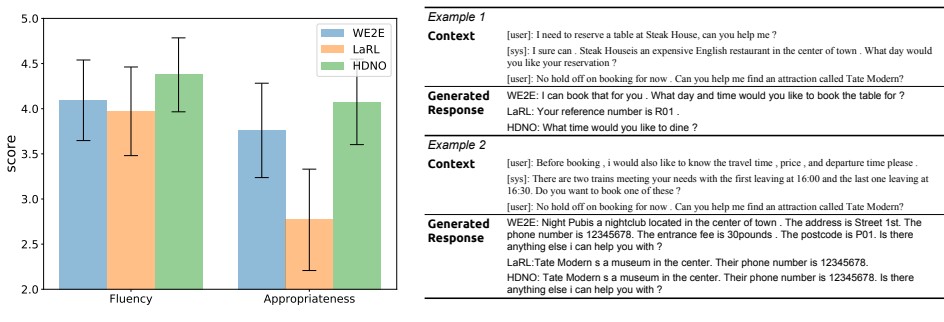

(a) Statistical results.

(b) Example generated responses.

Figure 3: (a) This figure shows the statistical results of human evaluation, where 30 people participated in this human evaluation and the questionnaire is constituted of 31 randomly selected turns of dialogues. (b) This figure shows the example generated responses from the questionnaire.

generated responses are (i.e., with no obvious grammar errors and redundant descriptions); and (2) appropriateness: how related the generated responses are to the provided context. For each criterion, a human assessor is allowed to give a score ranging from 1 to 5, where 5 indicates the best and 1 indicates the worst performance. Then, we calculated the mean and the variance for the score of each model. The final results and some example generated responses from the questionnaire are shown in Figure 3. The proposed HDNO performs best in the human evaluation, especially outperforming LaRL (Zhao et al., 2019) on appropriateness so much.

## B.2 LEARNING CURVES

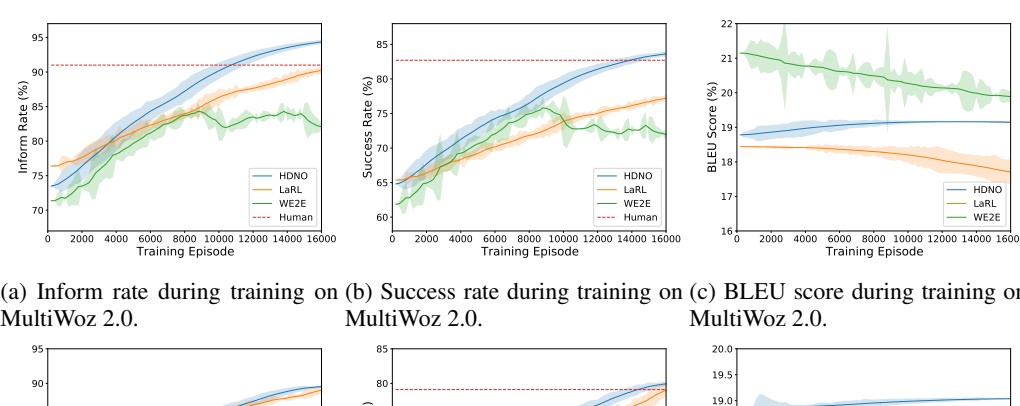

(a) Inform rate during training on MultiWoz 2.0.

(b) Success rate during training on MultiWoz 2.0.

(c) BLEU score during training on MultiWoz 2.0.

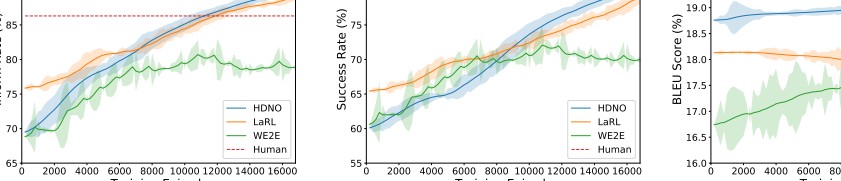

(d) Inform rate during training on MultiWoz 2.1.

(e) Success rate during training on MultiWoz 2.1.

(f) BLEU score during training on MultiWoz 2.1.

Figure 4: Validation inform rate, success rate and BLEU score during training.

In this section, we show the learning curves of HDNO (Async.) (i.e. abbreviated as HDNO in the rest of the appendix) and baselines on both MultiWoz 2.0 and MultiWoz 2.1. To show the performance of generalization, we only demonstrate the validation results during training. As seen from Figure 4, compared with baselines HDNO can preserve the comprehensibility (i.e. BLEU) while improving success rate and inform rate faster. We only show the results for the initial 16,000 episodes for conciseness of the figure.

### B.3    Complete Results with Beam Search

Table 3: The table shows the results of HDNO (Async.) with beam search on MultiWoz 2.0 and MultiWoz 2.1.

| Reward Shapes | $\alpha$ | Beam width = 1 | | | | Beam width = 2 | | | | Beam width = 5 | | | |
|---|---|---|---|---|---|---|---|---|---|---|---|---|---|
| | | Inform(%) | Success(%) | BLEU(%) | Total | Inform(%) | Success(%) | BLEU(%) | Total | Inform(%) | Success(%) | BLEU(%) | Total |
| | 0.0001 | 96.70 | 84.50 | 18.72 | 109.32 | 96.40 | 84.70 | 18.85 | 109.40 | 96.30 | 84.30 | 18.70 | 109.00 |
| MultiWoz 2.0 | 0.0005 | 96.30 | 83.60 | 18.60 | 108.55 | 96.40 | 84.10 | 18.30 | 108.61 | 96.30 | 84.90 | 18.50 | 109.10 |
| (Success + Discriminator) | 0.001 | 95.50 | 83.90 | 19.29 | 108.99 | 96.20 | 84.20 | 19.04 | 109.24 | 95.70 | 84.40 | 18.80 | 108.85 |
| | 0.005 | 97.00 | 84.10 | 18.72 | 109.27 | 96.50 | 84.30 | 18.78 | 109.18 | 96.80 | 83.90 | 18.70 | 109.05 |
| MultiWoz 2.0 (Success) | - | 95.70 | 82.00 | 18.83 | 107.68 | 96.10 | 84.20 | 18.51 | 108.66 | 95.30 | 82.70 | 18.57 | 107.57 |
| MultiWoz 2.0 (Success + BLEU) | - | 94.90 | 81.20 | 19.19 | 107.24 | 95.60 | 83.30 | 18.99 | 108.44 | 95.70 | 83.20 | 18.78 | 108.23 |
| MultiWoz 2.1 (Success + Discriminator) | 0.01 | 92.50 | 82.50 | 19.16 | 106.66 | 92.60 | 80.80 | 19.09 | 105.79 | 92.80 | 83.00 | 18.97 | 106.87 |

Table 4: The table shows the pretraining results with beam search on MultiWoz 2.0 and MultiWoz 2.1.

| | MultiWoz 2.0 | | | | MultiWoz 2.1 | | | |
|---|---|---|---|---|---|---|---|---|
| | Inform(%) | Success(%) | BLEU(%) | Total | Inform(%) | Success(%) | BLEU(%) | Total |
| Beam=1 | 69.50 | 62.00 | 19.10 | 84.85 | 70.70 | 61.40 | 18.24 | 84.29 |
| Beam=2 | 67.90 | 60.20 | 19.00 | 83.05 | 71.50 | 62.20 | 19.28 | 86.13 |
| Beam=5 | 69.50 | 62.50 | 19.20 | 85.20 | 71.40 | 62.80 | 19.12 | 86.22 |

Table 5: The table shows the results of HDNO (Sync.) with beam search on MultiWoz 2.0 and MultiWoz 2.1. The reward shape is the one we proposed in Section 5, with $\alpha = 0.0001$ for MultiWoz 2.0 and $\alpha = 0.01$ for MultiWoz 2.1 respectively.

| | MultiWoz 2.0 | | | | MultiWoz 2.1 | | | |
|---|---|---|---|---|---|---|---|---|
| | Inform(%) | Success(%) | BLEU(%) | Total | Inform(%) | Success(%) | BLEU(%) | Total |
| Beam=1 | 79.70 | 70.90 | 20.16 | 95.46 | 82.70 | 69.60 | 18.92 | 95.07 |
| Beam=2 | 79.60 | 70.80 | 19.71 | 94.91 | 81.40 | 69.40 | 19.17 | 94.57 |
| Beam=5 | 83.20 | 73.50 | 19.82 | 98.17 | 83.10 | 70.80 | 18.81 | 95.76 |

In this section, we show the complete results of HDNO (Async.) and HDNO (Sync.) as well as the pretraining results with beam search in Table 3, 5 and 4 respectively, where beam width is selected from 1, 2 and 5. Apparently, the proposed HRL algorithm gives an enormous improvement on the performance in comparison with that of the pretraining. Nevertheless, the pretraining is an essential part that cannot be replaced before training with reinforcement learning in the task-oriented dialogue system. The possibility of training from scratch in HRL with a good performance for the task-oriented dialogue system is left to be investigated.

### B.4    Complete Benchmark Results on MultiWoz 2.0

In this section, we show the complete state-of-the-art results on MultiWoz 2.0 for the policy optimization task, from the official records [4]. All these results are collected from their original papers. As Table 6 shows, HDNO leads the board on the total performance evaluated by $0.5 \times (\text{Inform} + \text{Success}) + \text{BLEU}$, where each metric is measured in percentage. However, due to the update of official evaluator this year, the results marked with * were probably underestimated.

### B.5    Examples of Generated Delexicalized Dialogues

In this section, we demonstrate some system utterances generated by the baselines and HDNO. Since most of the dialogues in MultiWoz 2.0 and MultiWoz 2.1 are similar, we only show the results on MultiWoz 2.0. As we can see from Table 7 and 8, compared with the baselines trained with SL (i.e. HDSA), the performance of HDNO on fulfilling a user's request is actually better, however,

---

[4]https://github.com/budzianowski/multiwoz.

Table 6: The table shows the full benchmark results on MultiWoz 2.0, compared with HDNO.

|  | Inform(%) | Success(%) | BLEU(%) | Total |
|---|---|---|---|---|
| HDNO (Our Model) | **96.40** | **84.70** | 18.85 | **109.4** |
| TokenMoE * (Pei et al., 2019a) | 75.30 | 59.70 | 16.81 | 84.31 |
| Baseline * (Budzianowski et al., 2018) | 71.29 | 60.96 | 18.80 | 84.93 |
| Structured Fusion * (Mehri et al., 2019) | 82.70 | 72.10 | 16.34 | 93.74 |
| LaRL * (Zhao et al., 2019) | 82.80 | 79.20 | 12.80 | 93.80 |
| SimpleTOD (Hosseini-Asl et al., 2020) | 88.90 | 67.10 | 16.90 | 94.90 |
| MoGNet (Pei et al., 2019b) | 85.30 | 73.30 | 20.13 | 99.43 |
| HDSA * (Chen et al., 2019) | 82.90 | 68.90 | **23.60** | 99.50 |
| ARDM (Wu et al., 2019) | 87.40 | 72.80 | 20.60 | 100.70 |
| DAMD (Zhang et al., 2019) | 89.20 | 77.90 | 18.60 | 102.15 |
| SOLOIST (Peng et al., 2020) | 89.60 | 79.30 | 18.30 | 102.75 |
| MarCo (Wang et al., 2020) | 92.30 | 78.60 | 20.02 | 105.47 |

the generated utterances of HDSA could be more fluent and comprehensible. In comparison with other baselines trained with RL, the generated utterances of HDNO is apparently more fluent and comprehensible. Especially, WE2E tends to generate as many slots as possible so as to increase the success rate, regardless of the comprehensibility of generated utterances. This is the common issue of most RL methods on task-oriented dialogue system as we stated in Section 3 in the main part of paper.

## C    EXTRA EXPERIMENTAL SETUPS

### C.1    TRAINING DETAILS

During pretraining, natural language generator (NLG) is trained via forcing the prediction at each time step to match an oracle word given a preceding oracle word as input. Nevertheless, during hierarchical reinforcement learning (HRL), NLG updates the estimated distribution at each time step by $\nabla J(\pi) = \mathbb{E}_\pi \left[ \sum_{i=t}^{T} \gamma^{i-t} r_i \nabla \ln \pi(\boldsymbol{w}_t | \boldsymbol{s}_t) \right]$ given a word sampled from a preceding predicted distribution. As for sampling a dialogue act from dialogue policy, we leverage reparameterization trick during pretraining. On the other hand, the discriminator is firstly pretrained and fixed as a reward during HRL.

We find that it is useful to train both of the discriminator and HDNO simultaneously during pretraining. Specifically, in addition to training on oracle system utterances, we also use the generated utterances to train the discriminator, which can improve its performance in experiments. The optimization problem is expressed as Eq.9, where $D(\cdot|\cdot, \theta)$ is the discriminator defined in Definition 2 in the main part of paper, parameterized with $\theta$; and $\eta$ is a multiplier to control the contribution of the generated utterances $(\hat{\boldsymbol{w}}_t)_{t=0}^{T'}$ given an initial state. We hypothesize that the generated utterances could expand the original small scale corpus (i.e. oracle system utterances), so as to improve the generalization of the discriminator. Since we have not grasped the intrinsic reason of this method, it is only regarded as a trick for training the discriminator in our work.

$$\max_\theta \sum_{t=0}^{T} \log D(\boldsymbol{w}_t | \boldsymbol{w}_{t-1}, \theta) + \eta \sum_{t=0}^{T'} \log D(\hat{\boldsymbol{w}}_t | \hat{\boldsymbol{w}}_{t-1}, \theta) \qquad (9)$$

During HRL, $r^{\text{succ}}$ and $r^{\text{disc}}$ are respectively normalized by z-score normalization (Patro and Sahu, 2015) to adaptively control their impacts in the total reward. For instance, when either $r^{\text{succ}}$ or $r^{\text{disc}}$ converges around some value, its normalized value will be close to zero and the total reward will be biased to the other. Therefore, this can further mitigate the conflicts between improving both the success rate and preserving the comprehensibility as stated in Section 3 in the main part of paper.

Furthermore, the experiment of HDNO was run on a Nvidia GeForce RTX 2080Ti graphic card, and it consumes around 3 hours for SL and 2 hours for RL. Therefore, it is not very expensive to reproduce our results.

Table 7: The table shows some delexicalized system utterances generated by the baselines and HDNO on MultiWoz 2.0.

| Models | Delexicalized Dialogue |
|---|---|
| HDNO (Our Model) | *usr*: i need train reservations from [value_place] to [value_place]
*sys*: what day would you like to travel ?
*usr*: i would like to leave on [value_day] and arrive by [value_time]
*sys*: [train_id] leaves at [value_time] and arrives at [value_time] . would you like me to book that for you ?
*usr*: before booking , i would also like to know the travel time , price , and departure time please .
*sys*: the travel time is [value_count] minutes . the price is [value_price] pounds . is there anything else i can help you with ?
*usr*: no hold off on booking for now . can you help me find an attraction called [attraction_name] ?
*sys*: [attraction_name] is a museum in the [value_area] . would you like their location ?
*usr*: yes , that was all i needed . thank you very much !
*sys*: you are welcome . have a nice day . goodbye . |
| WE2E | *usr*: i need train reservations from [value_place] to [value_place]
*sys*: i have [value_count] trains leaving [value_place] on [value_day] . what day and time would you like to travel ?
*usr*: i would like to leave on [value_day] and arrive by [value_time]
*sys*: the [train_id] arrives at [value_time] . the price is [value_price] pounds . the train id is [train_id] . the price is [value_price] pounds . the train id is [train_id] . the train id is [train_id] . the train id is [train_id] ...
*usr*: before booking , i would also like to know the travel time , price , and departure time please .
*sys*: the train id is [train_id] and the price is [value_price] pounds . the train id is [train_id] . the price is [value_price] pounds . the train id is [train_id] . the train id is [train_id] . the train id is [train_id] ...
*usr*: no hold off on booking for now . can you help me find an attraction called [attraction_name] ?
*sys*: [attraction_name] is a nightclub located in the [value_area] of town . the address is [attraction_address] . the phone number is [attraction_phone] . the entrance fee is [value_count] pounds . the postcode is [attraction_postcode] ...
*usr*: yes , that was all i needed . thank you very much !
*sys*: you are welcome . have a great day . goodbye . |
| LaRL | *usr*: i need train reservations from [value_place] to [value_place]
*sys*: i have a train leaving at [value_time] . would you like me to book that for you ?
*usr*: i would like to leave on [value_day] and arrive by [value_time]
*sys*: i have a train leaving at [value_time] . would you like me to book that for you ?
*usr*: before booking , i would also like to know the travel time , price , and departure time please .
*sys*: the [train_id] leaves at [value_time] and arrives at [value_time] . the travel time is [value_count] minutes.
*usr*: no hold off on booking for now . can you help me find an attraction called [attraction_name] ?
*sys*: [attraction_name] is located in the [value_area] of town . the address is [attraction_address] and the postcode is [attraction_postcode] .
*usr*: yes , that was all i needed . thank you very much !
*sys*: you are welcome . have a great day . |
| HDSA | *usr*: i need train reservations from [value_place] to [value_place]
*sys*: what day and time would you like to travel ?
*usr*: i would like to leave on [value_day] and arrive by [value_time]
*sys*: what time would you like to leave ?
*usr*: before booking , i would also like to know the travel time , price , and departure time please .
*sys*: the train leaves at [train_leaveat] and arrives by [train_arriveby] . the price is [train_price] pounds and the travel time is [value_count] minutes .
*usr*: no hold off on booking for now . can you help me find an attraction called [attraction_name] ?
*sys*: it is located in the [attraction_area] of town at [attraction_address] . their phone number is [attraction_phone] . is there anything else i can help you with ?
*usr*: yes , that was all i needed . thank you very much !
*sys*: you are welcome . have a great day ! |
| Human | *usr*: i need train reservations from [value_place] to [value_place]
*sys*: i have [value_count] trains matching your request . is there a specific day and time you would like to travel ?
*usr*: i would like to leave on [value_day] and arrive by [value_time]
*sys*: there are [value_count] trains for the day and time you request . would you like to book it now ?
*usr*: before booking , i would also like to know the travel time , price , and departure time please .
*sys*: there are [value_count] trains meeting your needs with the first leaving at [value_time] and the last [value_count] leaving at [value_time] . do you want to book [value_count] of these ?
*usr*: no hold off on booking for now . can you help me find an attraction called [attraction_name] ?
*sys*: yes it is a cinema located in the [value_area] part of town what information would you like on it ?
*usr*: thank you for using our system .
*sys*: you are welcome . have a nice day . goodbye . |

## C.2 HYPERPARAMETERS

The specific hyperparameters for training HDNO on MultiWoz 2.0 and MultiWoz 2.1 are shown in Table 9. During training, we used Adam optimizer (Kingma and Ba, 2015) for pretraining and stochastic gradient descent (SGD) for RL. To ease life, we explicitly label the hyperparameter if it is only for pretraining or RL.

## D EXTRA BACKGROUND

### D.1 REINFORCEMENT LEARNING AND POLICY GRADIENT METHOD

MDP (Bellman, 1957) is a discrete time stochastic control process that an agent decides an action $a_t \in \mathbb{A}$ at each state $s_t \in S$ emitted from an environment via a probabilistic transition function $p(s_{t+1}|s_t, a_t) \mapsto [0, 1]$. The environment also gives a reward $r(s_t, a_t, s_{t+1}) \mapsto \mathbb{R}$ (abbreviated

Table 8: The table shows some delexicalized system utterances generated by the baselines and HDNO on MultiWoz 2.0.

| Models | Delexicalized Dialogue |
|---|---|
| HDNO (Our Model) | *usr*: please help me make a taxi reservation
*sys*: what time would you like to leave ?
*usr*: it does not matter , just so i arrive by [value_time] .
*sys*: what is your destination ?
*usr*: i want to go to the [restaurant_name] . and i need the car type and contact number please .
*sys*: booking completed ! booked car type [taxi_type] contact number [taxi_phone] is there anything else i can help you with ?
*usr*: no that is all . thanks
*sys*: you are welcome . have a nice day . goodbye . |
| WE2E | *usr*: please help me make a taxi reservation
*sys*: i can help with that . what time do you need to leave or arrive by ?
*usr*: it does not matter , just so i arrive by [value_time] .
*sys*: i would be happy to help with your request , but i will need to know what time you are looking for .
*usr*: i want to go to the [restaurant_name] . and i need the car type and contact number please .
*sys*: i have booked a [taxi_type] for you . the contact number is [taxi_phone] . is there anything else i can help you with today ?
*usr*: no that is all . thanks
*sys*: you are welcome . have a great day . goodbye . |
| LaRL | *usr*: please help me make a taxi reservation
*sys*: i have booked a [taxi_type] for you . the contact number is [taxi_phone] .
*usr*: it does not matter , just so i arrive by [value_time] .
*sys*: i have booked a [taxi_type] for you . the contact number is [taxi_phone] .
*usr*: i want to go to the [restaurant_name] . and i need the car type and contact number please .
*sys*: i have booked a [taxi_type] for you . the contact number is [taxi_phone] .
*usr*: no that is all . thanks
*sys*: you are welcome . goodbye . |
| HDSA | *usr*: please help me make a taxi reservation
*sys*: what time would you like to leave ?
*usr*: it does not matter , just so i arrive by [value_time] .
*sys*: where will you be departing from and what is your destination ?
*usr*: i want to go to the [restaurant_name] . and i need the car type and contact number please .
*sys*: your taxi is booked . the car type is a [taxi_type] and the contact number is [taxi_phone] . is there anything else i can help you with ?
*usr*: no that is all . thanks
*sys*: you are welcome . have a great day ! |
| Human | *usr*: please help me make a taxi reservation
*sys*: i can help you with that . when do you need to leave ?
*usr*: it does not matter , just so i arrive by [value_time] .
*sys*: i can help you with that . where are you going ?
*usr*: i want to go to the [restaurant_name] . and i need the car type and contact number please .
*sys*: okay , i have a [taxi_type] for you with the contact number [taxi_phone] . is there anything else you need today ?
*usr*: no that is all . thanks
*sys*: you are welcome , have a wonderful time ! |

Table 9: Hyperparameters for training on MultiWoz 2.0 and MultiWoz 2.1.

| Hyperparameters | MultiWoz 2.0 | MultiWoz 2.1 | Description |
|---|---|---|---|
| max_utt_len | 50 | 50 | The maximum length of a user's utterances. |
| max_dec_len | 50 | 50 | The maximum length of system's utterances in a response. |
| batch_size (Pretrain) | 32 | 32 | The number of samples for each update during SL. |
| learning rate (Pretrain) | 1e-3 | 1e-3 | The learning rate for SL. |
| grad_clip (Pretrain) | 1.0 | 5.0 | The maximum total norm of gradients is allowed during SL. |
| dropout (Pretrain) | 0.5 | 0.5 | The randomness coefficient of dropout. |
| num_epoch (Pretrain) | 50 | 50 | The number of training epochs for SL. |
| embed_size | 100 | 100 | The size of a word embedding vector. |
| utt_cell_size | 300 | 300 | The size of hidden layer for utterance encoder. |
| dec_cell_size | 300 | 300 | The size of hidden layer for natural language generator. |
| y_size | 200 | 200 | The size of a latent dialogue act. |
| beta (Pretrain) | 1e-2 | 1e-3 | The regularization coefficient of KL divergence for dialogue policy during SL. |
| eta (Pretrain) | 0.1 | 0.1 | The coefficient to control the contribution of randomly generated utterances. |
| high-level learning rate (RL) | 9e-3 | 1e-2 | The learning rate for high-level policy during RL. |
| low-level learning rate (RL) | 9e-3 | 1e-2 | The learning rate for low-level policy during RL. |
| num_epoch (RL) | 1 | 2 | The number of training epochs for RL. |
| temperature (RL) | 0.1 | 0.1 | The temperature of a Gibbs distribution for each word during RL. |
| gamma (RL) | 0.99 | 0.99 | The discount_factor for the reward of success rate during RL. |
| gamma_nll (RL) | 0.99 | 0.99 | The discount_factor for the reward of nll generated from discriminator during RL. |
| grad_clip (RL) | 0.85 | 0.95 | The maximum total norm of gradients is allowed during RL. |
| alpha (RL) | 1e-4 | 1e-2 | The multiplier to adjust the balance between the discriminator and the success rate in the total reward during RL. |
| disc (Pretrain) | true | true | Pretraining discriminator. |
| gen_guide (Pretrain) | true | true | Using generated samples for pretraining discriminator. |
| reg (Pretrain) | kl | kl | Applying KL-divergence during pretraining. |
| high_freq (RL) | 1 | 1 | The frequency for updating high-level policy. |
| low_freq (RL) | 1 | 1 | The frequency for updating low-level policy. |
| synchron (RL) | false | false | Applying asynchronous updates between high-level and low-level policy during RL. |
| disc2reward (RL) | true | true | Applying discriminator as a reward. |
| success2reward (RL) | true | true | Applying success rate as a reward. |
| nll_normalize (RL) | true | true | Applying z-score normalization for nll generated from discriminator during RL. |

as $r_t$ for simplicity) to measure the performance of an action at each time step. RL (Sutton and Barto, 2018) is a learning paradigm which aims to find an optimal policy $\pi(\boldsymbol{a}_t|\boldsymbol{s}_t) \mapsto [0, 1]$ to deal with MDP by maximizing the expectation over cumulative long-term rewards. Mathematically, it can be expressed as $\max_\pi \mathbb{E}_\pi[\sum_{t=0}^\infty \gamma^t\, r_t]$, where $\gamma \in (0, 1)$ is a discount factor. Different from value-based methods, the policy gradient method derives a stochastic policy directly by optimizing a performance function w.r.t. parameters of the policy (Sutton and Barto, 2018). However, since the performance function cannot be differentiated w.r.t parameters of the policy, the policy gradient is derived by a natural gradient such that $\nabla_\theta J(\theta) = \mathbb{E}_\pi[\, Q(\boldsymbol{s}_t, \boldsymbol{a}_t)\nabla_\theta \ln \pi_\theta(\boldsymbol{a}_t|\boldsymbol{s}_t)\,]$. REINFORCE (Williams, 1992) is a policy gradient method that evaluates $Q(\boldsymbol{s}_t, \boldsymbol{a}_t)$ by a return $G_t = \sum_{t=0}^\infty \gamma^t\, r_t$. To deal with a continuous action space, $\pi_\theta(\boldsymbol{a}_t|\boldsymbol{s}_t)$ can be represented as a Gaussian distribution (Sutton and Barto, 2018), where the mean and scale are both parameterized with $\theta$.

## E    DIALOGUE ACT REPRESENTED AS ONTOLOGY

Table 10: Dialogue act ontology.

| **Dialogue Act Type** | inform / request / select / recommend / not found / request booking info / offer booking / inform booked / decline booking / welcome / greet / bye / reqmore |
| --- | --- |

In this section, we show the ontology for representing a handcrafted dialogue act (Budzianowski et al., 2018) in Table 10. The semantic meanings of latent dialogue acts analyzed in Section 5.2.3 in the main part of paper are correlated to these dialogue act types. This is the reason why we conclude that learning latent dialogue acts can potentially substitute for handcrafted dialogue acts with ontology.

## F    EXTRA DISCUSSION

**Discussion on Reinforcement Learning for Task-oriented Dialogue System:** As we stated in Section 3 in the main part of paper, the primary issue of using reinforcement learning (RL) in task-oriented dialogue system is that the improvement on fulfilling user requests and comprehensibility of generated system utterances is not simple to be balanced. One reason could be that the reward is only set up to improve the success rate and during learning the aspect of comprehensibility may be easily ignored. This is the reason why we consider extra reward criteria in our work to mitigate this predicament. Another reason could be that the state space and action space are so large in an end-to-end (E2E) model so that learning a mapping between these two spaces becomes difficult, as stated in Section 1 in the main part of paper. To deal with it, we propose to decouple dialogue policy and natural language generator (NLG) of an E2E model into two separate modules as those in the traditional architecture during learning, as well as model them with the option framework (Sutton et al., 1999). As a result, the complexity of mapping from context to system utterances is reduced from $V^2$ to $(L + M)V + ML$, where $V$ is the vocabulary size, $L \ll V$ is the space size of latent dialogue acts and $M \ll V$ is the space size of encoded utterances.

