# OpenReview forum: "Modelling Hierarchical Structure between Dialogue Policy and Natural Language Generator with Option Framework for Task-oriented Dialogue System"
_ICLR.cc/2021/Conference — ICLR 2021 Poster_

### Official Review · AnonReviewer2 · 2020-10-19
**Tend to Accept**

**Rating:** 6
**Confidence:** 4

**Review:**

Summary:
This paper proposes modeling the hierarchical structure between dialog policy and natural language generator with option network and train it with HRL. It also introduces a discriminator modeled with language models as an additional reward, which further improves the learning procedure's comprehensibility. Besides, this paper has demonstrated the interpretability of the latent dialog act via clustering methods.

Pros:
1. This paper proposed formulating dialog policy as a high-level policy over dialog act and NLG as a low-level policy and train the word policy module using HRL. It is another move towards improving policy learning via HRL.
2. The proposed model achieves SOTA performance in the policy optimization leaderboard on the MultiWOZ dataset.
3. It is interesting to see that when formulated as latent factors, the distribution of dialog acts still shows a clustering pattern, which manifests the model's interpretability.
4. This paper introduced an additional reward using a discriminator of language models. The ablation study shows that this reward can be useful (though not significant).

Cons:

The proposed HRL approach is a direct application of the option framework on a task-oriented dialog system. It is similar to the hierarchical structure in the open-domain dialog system.

Minor issues:
1. Introduction paragraph 2: When introducing end2end models, there is a lack of citations of recent e2e models including DAMD, SimpleTOD, SOLOIST, etc.
2. Abstract Line 9:  o gur -> our

---

> ### Author Response · Authors · 2020-11-13
> **Response to R2**
>
> Dear R2,
>
>
> Thank you for your comments. We will respond to your concerns in detail.
>
> Q1. "The proposed HRL approach is a direct application of the option framework on a task-oriented dialog system. It is similar to the hierarchical structure in the open-domain dialog system."
>
> A1. We don't agree that our proposed approach is a direct application of the option framework. The stage of rational modelling the task oriented dialogue system is fundamental to apply the option framework. Although the hierarchical structure is similar to the one in the open-domain system, the underlying purposes of each module (or hierarchies) are different. In a task-oriented system, it started from modular frameworks, where each module plays a specific role (see the new addition in Section Background in the revised paper). However, due to the difficulty to capture the labels and the noisy annotations for the dataset, researchers start finding solutions to avoid using these labels during learning. Our work follows this idea, ignoring the dialogue act labels and designing a learning process (HRL based on the option framework) so that the dialogue policy and NLG can learn well. On the other hand, an open-domain dialogue system initially concentrated on direct mapping between user dialogues to the response. However, for easy learning by variational inference, they split the framework, like hierarchical, but the intrinsic purpose is the encoder and decoder for VAE. For the reasons stated above, we don't agree this is a con.
>
> Q2. "This paper introduced an additional reward using a discriminator of language models. The ablation study shows that this reward can be useful (though not significant)"
>
> A2. We did the ablation study for our proposed additional reward using a discriminator of language models. Unfortunately, due to the limitation of the space in the initial submission, we leave it in the Appendix B.2, however, we mention it in the main part of paper (see the last line in Section 6.2.1). To avoid the misunderstanding, we will place it in the main part of paper in the revised version of paper thanks to the 9 pages regulations.
>
>
> Finally, we thank you for your time in the comments again. We are glad to have an academic discussion here.
>
>
> Best Regards,
>
> Authors of paper 443

---

### Official Review · AnonReviewer4 · 2020-10-27
**interesting work but the dialog task is basic.**

**Rating:** 6
**Confidence:** 4

**Review:**

This paper attempts to model task-oriented dialogue system using hierarchical reinforcement learning between the actions policy and natural language generation system. The utterances are encoded with GRU cells, the action policy is one-layer linear model and outputs the mean and variance of a multivariate Gaussian distribution, and the NLG is an LSTM decoder.

The paper is well written, easy to follow, and adequately motivated.

Pros:

Hierarchical RL formulation with options for joint action policy and natural language generation

The paper proposes asynchronous updates between dialogue policy and NLG to theoretically guarantee their convergence to a local maximizer.

It also proposes using a language models as a discriminator model for reward assignment to further improve the comprehensibility of generated responses.

Pre-training of task-oriented dialog models with  variational bayes similar to VHRED (Serban et al., 2017).

Experiments show significant performance improvement over the baselines with both automatic and human evaluations.

Also, the results show that continuous representation of the action policy in HDNO performs better than the discrete representation in the LaRL, which is very interesting and informative.

Cons:
The use of an oracle dialogue state and an oracle database search result. This is a major drawback of this work. Recent work in this space are now considering imperfect dialogue state and the corresponding database search results. There is also an increasing need to consider non-trivial database interactions such as booking for more practical applications.

The evaluation/comparison using updated official evaluator may be missing for some of the more recent work e.g..,

GPT-2 (Budzianowski and Vulic 2019)
Structured Fusion (Mehri, Srinivasan, and Eskenazi 2019) SOLOIST (Peng et al. 2020)
DSTC8 Track 1 Winner (Ham et al. 2020)
SimpleTOD (Hosseini-Asl et al. 2020)


Questions:

What do you mean by “Distinguished from a conventional modular system, we additionally give a context to NLG to satisfy the option framework.”? You mean the dialogue context is equivalent to the state space in HDNO formulation?

How are the oracle dialogue state and an oracle database search results encoded?

What kind of model is used for the language model discriminator? LSTM?

How did you handle the action policy toward the database for booking, which can cause database mutation? Are those ignored?

Are the results in Table 5 based on the updated official evaluator similar to Table 1? If not, then the comparison in Table 5 is not apples to apples.

---

> ### Author Response · Authors · 2020-11-13
> **Response to R4 (Part 1)**
>
> Dear R4,
>
>
> Thank you for your time in the comments. Next, we will respond to your concerns and questions specifically.
>
> Q1. "What do you mean by “Distinguished from a conventional modular system, we additionally give a context to NLG to satisfy the option framework.”? You mean the dialogue context is equivalent to the state space in HDNO formulation?"
>
> A1. The dialogue context plays the role of the state that an option is activated. The space of dialogue contexts is equivalent to the state space where an option can start. In the option framework, if an option is activated, the low level policy belonging to this option will be used for the next several interactions to the environment until it is terminated [1]. In HDNO, NLG plays the role of the low level policy that is controlled by an option (i.e. a latent dialogue act), i.e., different dialogue acts should induce different NLGs. Moreover, the low level policy (i.e. NLG in HDNO) should do the first action according to the state that activates the option. This is the reason why we give additional dialogue contexts to the low level policy when it determines the first word. However, in the conventional task-oriented dialogue system, only dialogue act is given to the NLG to generate the utterances. In conclusion, we do this just for implementing the statement in the definition of HDNO, rigorously following the theoretical setup of the option framework, reducing the gap between theory and implementation.
>
> Q2. “How are the oracle dialogue state and an oracle database search results encoded?”
>
> A2.  For this question, you can refer to the MultiWoz paper [2] to see more details. In our paper, this is not a critical point, we just use the official oracle dialogue act and database search results provided by the MultiWoz datasets. Nevertheless, we actually use a linear layer + ReLU to transform the concatenation of oracle dialogue state, oracle database search and latent dialogue acts to a vector with the same size as the hidden state of LSTMs.
>
> Q3. “What kind of model is used for the language model discriminator? LSTM?”
>
> A3. Yes, in our implementations, the language model discriminator is implemented as RNN with LSTMs. We may forget to illustrate it in the main part of the paper, but we add it in Section of introducing the language model discriminator in the revised paper. However, it can be implemented with other network structures. It is interesting to investigate it in the future work.
>
> Q4. “How did you handle the action policy toward the database for booking, which can cause database mutation? Are those ignored?”
>
> A4. This is not the perspective of our work. In our work, we proposed a general learning framework for solving comprehensive tasks for policy optimization. However, the point you mentioned is possible to be extended from our framework in the future work.
>
> Q5. “Are the results in Table 5 based on the updated official evaluator similar to Table 1? If not, then the comparison in Table 5 is not apples to apples.”
>
> A5. The results shown in Table 5 are the official records (see https://github.com/budzianowski/multiwoz). As mentioned below the table Policy Optimization, only LaRL, Structured Fusion, Baseline (i.e. the one we used for e2e RL in our work), TokenMoE and HDSA reported the results on the old official evaluator. In other words, most of the other results are evaluated via the updated official evaluator. To fairly compare the results of the methods highly related to our work, we re-run these algorithms (i.e. LaRL, HDSA and BASELINE trained with e2e RL called WE2E in our paper) by the original source codes. For this reason, we think most comparisons shown in Table 5 are apples to apples. To remove the confusion, we add additional annotations to Table 5 in the revised paper.

---

> ### Author Response · Authors · 2020-11-15
> **Response to R4 (Part 2)**
>
> Q6. "The use of an oracle dialogue state and an oracle database search result. This is a major drawback of this work. Recent work in this space are now considering imperfect dialogue state and the corresponding database search results. There is also an increasing need to consider non-trivial database interactions such as booking for more practical applications."
>
> A6. We argue that the task-oriented dialogue systems can be modular. If saying dialogue state tracker and database searcher are two modules (i.e. two front-end modules), our work solves the modules dialogue policy and natural language generator (i.e. two backend modules). For this reason, our proposed method can be extended to the scenario you mentioned. Therefore, we don't think this is a drawback, which is just the different perspective of research. In conclusion, we have stated that we will consider all modules in the future work.
>
>
> Finally, thank you for your comments.
>
>
> Best Regards,
>
> Authors of paper 443
>
> Reference
>
> [1] Richard S Sutton,  Doina Precup,  and Satinder Singh.   Between mdps and semi-mdps:  A frame-work for temporal abstraction in reinforcement learning.Artificial intelligence, 112(1-2):181–211, 1999.
>
> [2] Pawel Budzianowski, Tsung-Hsien Wen, Bo-Hsiang Tseng, I ̃nigo Casanueva, Stefan Ultes, OsmanRamadan, and Milica Gasic. Multiwoz - A large-scale multi-domain wizard-of-oz dataset for task-oriented dialogue modelling.  InProceedings of the 2018 Conference on Empirical Methods inNatural Language Processing, Brussels, Belgium, October 31 - November 4, 2018, pages 5016–5026. Association for Computational Linguistics, 2018.
>
> [3] Tiancheng Zhao, Kaige Xie, and Maxine Eskenazi.Rethinking action spaces for reinforce-ment learning in end-to-end dialog agents with latent variable models.arXiv preprintarXiv:1902.08858, 2019.

---

> ### Comment · AnonReviewer4 · 2020-11-18
> **Good response to most questions but still disagree on the value of the task**
>
> Thank you for your detailed response to the questions asked, especially with the updates to Table 6. Unfortunately, I still disagree with your response to question Q6. Although the system is modular, the POL and NLG components are downstream, and therefore, would be more affected by errors propagating from the upstream state tracker and database modules. For example, the approach in [1] total ignores the POL module and still shows a competitive performance. This is why the task not exploring the impact of imperfect belief state and db results on the system performance is limited. It is possible that the proposed framework may not be as robust to upstream errors compared to existing work.
>
> For this reason, I would keep my score as it is.
>
> [1] Baolin Peng, Chunyuan Li, Jinchao Li, Shahin Shayandeh, Lars Liden, and Jianfeng Gao. Soloist: Few-shot task-oriented dialog with a single pre-trained auto-regressive model. arXiv preprint arXiv:2005.05298, 2020.

---

> > ### Author Response · Authors · 2020-11-23
> > **Response to R4 (Stage 2)**
> >
> > Dear R4,
> >
> >
> > First, we agree with this statement that “Although the system is modular, the POL and NLG components are downstream, and therefore, would be more affected by errors propagating from the upstream state tracker and database modules”. However, in [1] the downstream module ignoring POL (replacing POL and NLG by one response generator) still cannot avoid the impact of imperfect belief state and db results. Second, this task is valuable, even though the method in [1] trained with the belief state tracker (if we hypothesize the full stack training can benefit the modules of the whole system), its performance on the context-to-response task (i.e. the task we considered in our paper) still cannot perform better than that of our method. This means studying how to exclusively train the downstream modules is valuable. Nevertheless, we agree that considering the upstream modules is necessary, which is our future work (see conclusion). Besides, this is not conflicted with the task we considered in our paper. In our view, the performance of the whole system can be additively improved when considering more modules.
> >
> > Whatever, we respect your opinions. Thank you for the meaningful academic discussion here.
> >
> >
> > Best Regards,
> >
> > Authors of paper 443
> >
> > Reference
> >
> > [1] Baolin Peng, Chunyuan Li, Jinchao Li, Shahin Shayandeh, Lars Liden, and Jianfeng Gao. Soloist: Few-shot task-oriented dialog with a single pre-trained auto-regressive model. arXiv preprint arXiv:2005.05298, 2020.

---

### Official Review · AnonReviewer3 · 2020-10-30
**More discussion/ablation on the proposed method study required.**

**Rating:** 6
**Confidence:** 4

**Review:**

Summary
The paper looks the problem of lack of comprehensibility that arises when we use RL to train a E2E dialog system to maximise a given reward function. The paper proposes a  HRL/options framework based method to learn a dialog policy over learned latent dialog acts which can then guide the lower level NLG. This along with a regularization reward using language model the paper aims to improve comprehensibility. The show improved performance in MultiWoz dataset.

Strengths
1. The paper looks at a very relevant problem in dialog research. The ability to use RL along with SL and being able to use RL without compromising on comprehensibility.
2. The paper proposes Options framework based method for using the hierarchical structure in dialog to learn the dialog policy and NLG in a hierarchical fashion.
3. They provide a training that guarantees convergence to local maxima.
4. They show improved total performance in MultiWoz dataset compared to recent, relevant baselines.


Weakness/Comments/Questions
1. The paper starts with the motivation of handling comprehensibility. More discussion is required on a) what are the reasons of loss in comprehensibility in this case (it is briefly mentioned in the intro) b) why their individual design choices and how they handles the different reasons c) some evaluation to verify this
2. It would helpful to place it more clearly where the contribution of the paper lies in the related work. My understanding is that contribution of the paper is in exploring using options framework to goal-oriented dialog to handle the issue in question. HRL in general has been used previously for goal-oriented dialog, using language models to regularize RL models has been used and pertaining using SL is widely used. If there are particular differences in the above, it would nice to clearly state them and also say why the different choices and verify if the different choices are beneficial compared to the previous ones.
3. Relevant to some of the above points. It is not clear to me why we cannot use HDSA+R or LARL + NLG + language model reward. Not necessarily these particular combinations. Some discussion on the current design choices and why making the proposed methods features to some of the other baselines is not a way to achieve some benefits is not the right way to do it. It's ok for the proposed method to be one particular way, but that discussion would be useful.
4. Clarification on the task setting: Is it the case that the agent's current utterance does not decide what the next user utterance is? i,.e the agent is given the ground truth context every time and asked to predict the correct next utterance. That prediction does not affect the way in which the overall dialog goes? If that is the case, that should be made more explicit. In that case, the dialog policy learned is more of a contextual bandit setting. The complexity of learning options would be way different in the two different settings. Just wanted to understand it.
5. In the 6.2.3 visualization of clusters, it would be very useful to have a visualization of clusters from some baselines on other ways of learning. To see if this is due to some new addition by the paper or is generally present.
6. There are several parts to the method and there are I assume several differences in the architecture etc with baselines etc. It would be really useful to have an ablation study to disentangle which piece of the training or method is contributing to what and how in the performance measure. Otherwise for example it is not clear to me if the improvement in Blue compared to LaRL comes from the extra reward using the language model or from the options framework. What happens when the extra reward for using the language model is added to LaRL (that might be tough if you have to modify others code). what part of the performance is coming from pretraining (especially if using VAE type is novel, then quantifying that is important with and without VAE type SL), etc.

Questions to authors
1. It would be great if you could respond to some of the above comments. Thanks.

Minor
1. Typo in the abstract: *In our work, we
2. Move to E2E system can be motivated a bit more (allows end-user feedback to be passed through all modules easily and don't have to worry about how a change in one module affects all other modules explicitly etc)
3. Might be useful to define what is exactly meant by 'comprehensibility'
4. The intro says "the goal is absolutely clear". Not sure if that is really true. For example, it's not clear what the correct reward function to provide to the RL agent is. As the paper points out, the success rate alone is not enough. The dialog needs to be polite, follow natural language, short, etc which are hard to automatically measure. Human evaluation is costly and could also have bias like the paper points out. Just trying to say that automatic evaluation of dialog systems is a hard problem.
5. I am just curious why you use GRU for the encoder and LSTM for the decoder?

After author response
The authors have responded to most of my questions/concerns satisfactorily.
Changing my score from 4 to 6

---

> ### Author Response · Authors · 2020-11-13
> **Response to R3 (Part 1)**
>
> Dear R3,
>
>
> Thank you for your so detailed comments. We will address some of your questions and concerns.
>
> Q2. "It would helpful to place it more clearly where the contribution of the paper lies in the related work. My understanding is that contribution of the paper is in exploring using options framework to goal-oriented dialog to handle the issue in question. HRL in general has been used previously for goal-oriented dialog, using language models to regularize RL models has been used and pertaining using SL is widely used. If there are particular differences in the above, it would nice to clearly state them and also say why the different choices and verify if the different choices are beneficial compared to the previous ones."
>
> A2. Firstly, we would like to argue that by the side of applied science such as NLP, the motivation of applying some theoretical framework is very important. HRL has been actually applied to the task-oriented dialogue systems before, however, these works mainly focus on only modelling **the dialogue policy as a hierarchical structure** and applying the HRL for only training dialogue policy.  In contrast, our work focuses on **the hierarchical structure between dialogue policy and natural language generator** and applying the option framework for training on this hierarchical structure. As for using language models, the motivation is solving the problem of sparse reward in the task-oriented dialogue systems. From this perspective, we believe that we are the first one to do it in this field. Moreover, we never claimed that pretraining is a novelty. As your requests, we improve our related work in the revised paper, and claim our contributions more clearly on the part of HRL for the hierarchical structure between dialogue policy and NLG.
>
> Q3. "Relevant to some of the above points. It is not clear to me why we cannot use HDSA+R or LARL + NLG + language model reward. Not necessarily these particular combinations. Some discussion on the current design choices and why making the proposed methods features to some of the other baselines is not a way to achieve some benefits is not the right way to do it. It's ok for the proposed method to be one particular way, but that discussion would be useful."
>
> A3. Firstly, in the paper of LaRL, they claimed that applying RL on training dialogue policy only with fixing NLG can result in good performance, which is one of the novelties of their work. However, in our work, we actually break out what they claimed in their paper by training both dialogue policy and NLG. To make the training with some theoretical guarantee, we propose applying the option framework on modelling the relation between dialogue policy and NLG. Therefore, our work can be seen as one extension from LaRL that we mentioned in the related work (see the last sentence in the first paragraph of related work). In the implementation, we try to **obey the setup in the option framework** so that the theoretical tool can be used as explaining the resultant learning process. We think this is also useful for the analysis in the follow-up work.  As for HDSA+R, we don't think this is necessary for us to consider. As stated above, the motivation of using discriminator modelled with language models as an auxiliary reward is for solving the problem of sparse reward in RL. From this perspective, since HDSA is originally trained by SL rather than RL algorithms, it is meaningless to think of it in our paper, however, we think it could be interesting to investigate this in future work. In addition, since HDSA possesses so complicated architecture, especially the dialogue act being represented by a hierarchical structure of semantic representations, e.g. ontology [2], how to train it by RL is an open question. Meanwhile, the dialogue act and NLG in HDSA are separately trained with labels, which is also a challenge to directly train it with RL.
>
> Q5. “In the 6.2.3 visualization of clusters, it would be very useful to have a visualization of clusters from some baselines on other ways of learning. To see if this is due to some new addition by the paper or is generally present.”
>
> A5. Yes, we agree with this. We will add the visualization of clusters of latent dialogue acts for LaRL, since WE2E has no latent dialogue acts and the dialogue act in HDSA is directly the ontology. The figure will be uploaded to the supplementary materials in rebuttal revision. From the clustering figure of LaRL, we cannot explicitly see any explainable semantic meanings similar to that of HDNO. Therefore, we are confident to say the resultant ability of explanation is due to the new addition from our paper.

---

> ### Author Response · Authors · 2020-11-15
> **Rebuttal to R3 (Part 2)**
>
> Q4. "Clarification on the task setting: Is it the case that the agent's current utterance does not decide what the next user utterance is? i,.e the agent is given the ground truth context every time and asked to predict the correct next utterance. That prediction does not affect the way in which the overall dialog goes? If that is the case, that should be made more explicit. In that case, the dialog policy learned is more of a contextual bandit setting. The complexity of learning options would be way different in the two different settings. Just wanted to understand it."
>
> A4. In the paper of LaRL [1], they claimed that the task in the experiment is contextual bandits. From another perspective, this process for this specific experiment setting can be seen as a system with **p(s' | s)** and **r(s, s', a)**, i.e. MDP with actions independent transition setting [3]. The main difference from the original MDP is that the state will not be affected by the utterances, however, the reward will still be affected, since the success rate will only be given at the end of a session of dialogues according to the responses for each turn in this session. Our analysis is only the first step, and more complicated conditions can be taken into consideration in the future work. We would like to argue that our proposal is focusing on solving the general MDP, however, the existing user simulator is quite toy and biased, so the experimental results on these are not convincing. For this reason, we used the state-of-the-art benchmark MultiWoz for testing our algorithms.
>
> Q6. "There are several parts to the method and there are I assume several differences in the architecture etc with baselines etc. It would be really useful to have an ablation study to disentangle which piece of the training or method is contributing to what and how in the performance measure. Otherwise for example it is not clear to me if the improvement in Blue compared to LaRL comes from the extra reward using the language model or from the options framework. What happens when the extra reward for using the language model is added to LaRL (that might be tough if you have to modify others code). what part of the performance is coming from pretraining (especially if using VAE type is novel, then quantifying that is important with and without VAE type SL), etc."
>
> A6. Yes, we agree that the ablation study should be done and we actually did that. Due to the limitation of the main part of the paper, we left the evaluation of VAE type training and the ablation study on reward shaping (though on HDNO rather than LaRL) to the appendix (see Table 2 and Table 4 in Appendix). Thanks to one extra page in the rebuttal stage, we move these results to the main part of the paper in the revised paper to avoid misunderstanding. In addition, we will add one more ablation study to demonstrate the performance of the asynchronous update we proposed.  To ease your life, we list the ablation study results below.
>
> The results for reward shaping on MultiWoz 2.0:
>
> |                          | \\alpha | Inform \(%\) | Success \(%\) | BLEU \(%\) | Total   |
> |--------------------------|---------|--------------|---------------|------------|---------|
> | Success                  | \\      | 96\.10       | 84\.20        | 18\.51     | 108\.66 |
> | Success \+ BLEU          | \\      | 95\.60       | 83\.30        | 18\.99     | 108\.44 |
> | Success \+ Discriminator | 0\.0001 | 96\.40       | 84\.70        | 18\.85     | 109\.40 |
> |                          | 0\.0005 | 96\.30       | 84\.90        | 18\.50     | 109\.10 |
> |                          | 0\.001  | 96\.20       | 84\.20        | 19\.04     | 109\.24 |
> |                          | 0\.005  | 97\.00       | 84\.10        | 18\.72     | 109\.27 |
>
> The results for pretraining, synchronous updates and asynchronous updates for MultiWoz 2.0:
>
> |                      | Inform \(%\) | Success \(%\) | BLEU \(%\) | Total   |
> |------------------|-----------------|------------------|--------------|------|
> | Pretraining (Bayesian) | 69\.50 | 62\.00 | 19\.10 | 84\.85 |
> | HDNO (Async.) | 96\.40       | 84\.70        | 18\.85     | 109\.40 |
> | HDNO (Sync.)   | 83\.20       | 73\.50        | 19\.82     | 98\.17   |
>
> The results for pretraining, synchronous updates and asynchronous updates for MultiWoz 2.1:
>
> |                      | Inform \(%\) | Success \(%\) | BLEU \(%\) | Total   |
> |------------------|-----------------|------------------|--------------|------|
> | Pretraining (Bayesian) | 71\.40 | 62\.80 | 19\.12 | 86\.22 |
> | HDNO (Async.)            | 92\.80 | 83\.00 | 18\.97 | 106\.77 |
> | HDNO (Sync.)              | 83\.10       | 70\.80        | 18\.81     |  95\.76   |
>
> The results of HDNO (Sync.) and HDNO (Async.) in the latter two tables are obtained with the identical settings. We think these ablation studies are sufficient to support our contributions.

---

> ### Author Response · Authors · 2020-11-15
> **Rebuttal to R3 (Part 3)**
>
> Q7. “The intro says "the goal is absolutely clear". Not sure if that is really true. For example, it's not clear what the correct reward function to provide to the RL agent is. As the paper points out, the success rate alone is not enough. The dialog needs to be polite, follow natural language, short, etc which are hard to automatically measure. Human evaluation is costly and could also have bias like the paper points out. Just trying to say that automatic evaluation of dialog systems is a hard problem.”
>
> A7. We’d like to argue that there are two sorts of dialogue systems: task-oriented dialogue systems and open-domain dialogue systems. The one you know may be the open-domain dialogue system, whose primary objective is solely chatting with human beings. For this system, we agree that the goal is unclear, so many researchers in that area dedicate designing reasonable rewards for learning. However, what we consider in this paper is called the task-oriented dialogue system, whose primary objective is assisting completing a task through interacting with human beings. Therefore, the goal for this system is clear, i.e., completing the task or not. Nevertheless, it‘s better to consider the other effects besides the goal, e.g. the length of dialogues and responses, politeness of responses and so on. Standing by this side, we agree that these need to be explored in future. To make most audiences from various regions understand this problem well, we additionally add background about task-oriented dialogue systems in revised paper.
>
> Q8. “Move to E2E system can be motivated a bit more (allows end-user feedback to be passed through all modules easily and don't have to worry about how a change in one module affects all other modules explicitly etc)”
>
> A8. We would like to argue that we mention the drawbacks of E2E systems in the introduction, but we are happy to illustrate it here again. From the explanation of  [4] the state space and action space (represented as a vocabulary) in E2E fashion is so huge that learning to generate comprehensible utterances becomes difficult. From our own view, dialogue systems in E2E fashion may lack explanation during the procedure of decision. Nowadays, the ability of explanation is more and more important in machine learning, the trend is gradually changing from pursuing powerful performance to understanding the machine. These are the reasons why we don’t consider the E2E system in this paper.
>
> Q9. “I am just curious why you use GRU for the encoder and LSTM for the decoder?”
>
> A9. The main reason is the RL baselines, i.e. WE2E and LaRL use GRU for the encoder and LSTM for the decoder. In our view, it is fair to compare our method with the important baselines with the same architectures to show the power of our contributions.
>
> Thanks for your time on so specific and insightful comments again. We wish that you can reconsider your decision from the perspective of NLP after reading our explanations on your misunderstandings as well as the answers to your questions and concerns.
>
>
> Best Regards,
>
> Authors of paper 443
>
> Reference
>
> [1] Tiancheng Zhao, Kaige Xie, and Maxine Eskenazi.Rethinking action spaces for reinforce-ment learning in end-to-end dialog agents with latent variable models.arXiv preprintarXiv:1902.08858, 2019.
>
> [2] Wenhu Chen, Jianshu Chen, Pengda Qin, Xifeng Yan, and William Yang Wang. Semantically conditioned dialog response generation via hierarchical disentangled self-attention. In Proceedings of the 57th Conference of the Association for Computational Linguistics, ACL 2019, Florence, Italy, July 28- August 2, 2019, Volume 1: Long Papers, pages 3696–3709. Association for Computational Linguistics, 2019.
>
> [3] Potters, J. A., Raghavan, T. E. S., & Tijs, S. H. (2009). Pure equilibrium strategies for stochastic games via potential functions. In Advances in Dynamic Games and Their Applications (pp. 1-12). Birkhäuser Boston.
>
> [4] Lewis, M., Yarats, D., Dauphin, Y. N., Parikh, D., & Batra, D. (2017). Deal or no deal? end-to-end learning for negotiation dialogues. arXiv preprint arXiv:1706.05125.

---

> ### Comment · AnonReviewer3 · 2020-11-18
> **Satisfactory response to most questions**
>
> Thanks for the response. I am happy with the response to most of the questions. Changing my score from 4 to 6.
> Will go over the revised submission when available and see if any further changes are required in the score.

---

> > ### Author Response · Authors · 2020-11-18
> > **Thank you for upgrading the score**
> >
> > Dear R3,
> >
> >
> > Thank you for upgrading the score. We think the revised paper is available now and you can download it by clicking the icon "PDF" beside the title.
> >
> >
> > Best Regards,
> >
> > Authors of paper 443

---

### Official Review · AnonReviewer1 · 2020-11-01
**Good paper with both theoretical and empirical contributions.**

**Rating:** 7
**Confidence:** 4

**Review:**

Summary
=========
Authors applied reinforcement learning framework to the problem of task-oriented dialog. In particular, they used the option framework to represent the connection between the dialog policy and the natural language generation. Theoretically, they showed that synchronized updates to the low-level and high-level policy may never converge, yet asynchronized updates guarantees convergence. Authors also used a discriminator reward signal to cope with sparse reward (dialog success rate) and better representation of the human evaluation.

Empirical results on MultiWoz 2 and 2.1 shows improvement over other state-of-the-art techniques.

Summary:
+ Appealing theoretical contributions
+ Empirical results are encouraging
+ The use of discriminator for reward shaping in addition to task success rate is interesting
- Writing and explanation can be improved. There are few places (see details) that authors have assumptions in mind but do not provide those assumptions until later. Would be great to state them upfront to avoid confusion.
- The original option framework assumes given options. Given the ending of the paper, I interpreted that the set of dialog acts (i.e. options) are learnt automatically but I could not find this to be communicated explicitly.

Questions:
Did you look into the quality of dialogs specifically? In some of our recent experiments we found out that those automatic metrics do not necessarily correspond to great user experience.

Details
=========
Abstract: In o gur work => In our work
P5: "oracle dialogue state": What is the oracle dialog state and how is it calculated? The world oracle conveys the meaning of being absolute truth which sounds a bit unexpected. The same comment is applicable to oracle database (DB) results. Is it possible in your system you queried the DB with wrong parameter? If yes, do you still name the output of the DB as oracle results? => Ah. you explained this in page 6, in Task Description. I highly recommend bringing this assumption earlier to avoid readers confusion.
P5: Section 4.4: I am still eager to know how you select your dialog actions. Would be great to tell your reader earlier.

---

> ### Author Response · Authors · 2020-11-13
> **Response to R1**
>
> Dear R1,
>
>
> Thank you for your insightful comments. We will respond to your comments and answer the questions that you raised.
>
> Firstly,  we agree that some assumptions (e.g. the oracle DB, the oracle dialogue state) can be mentioned earlier, and we will do that in the revised version of paper.
>
> Secondly, we now answer the questions you raised.
>
> Q1. "P5: Section 4.4: I am still eager to know how you select your dialog actions."
>
> A1. Similar to the LaRL [1], we use a latent vector with continuous multivariables and we assume that this latent vector is generated from the dialogue policy (i.e. an encoder in the framework of VAE) follows the isotropic multivariable Gaussian distribution. In Section 4.4, during the pretraining with a variational lower bound, we sampled each latent vector from the dialogue policy with reparameterization trick so that the backpropagation can work in training. The sampled latent vector is each sampled dialogue act, which means we did not use any dialogue act label.
>
> Q2. "Did you look into the quality of dialogs specifically? In some of our recent experiments we found out that those automatic metrics do not necessarily correspond to great user experience."
>
> A2. Yes, we looked into the quality of dialogues specifically. We agree that these automatic metrics do not necessarily correspond to the great user experience. However, we would like to claim that the automatic metrics can somewhat reflect the general quality of dialogues, though it is difficult to distinguish that affected by user preferences. In our view, the main underlying factor causing the problem you mentioned is the diversity of user minds, e.g., for the same dialogue, different users could have different comments on it. Unfortunately, it is difficult to be addressed at the moment, since the study on user models is not enough up to now. One possible solution is motivating the study on user models, since if we understand users well then we can design a more rational automatic metric. On the other hand, the study on user models can also benefit the modelling of user models, so that we can use more friendly and realistic user simulators for training RL algorithms on task-oriented dialogue systems. Most of the existing user simulators are quite "toys", and this forces us to use the dataset collected from real users for training RL algorithms which may slightly violate the setups of RL.
> Although human evaluation could mitigate the problems of existing automatic metrics, it is quite subjective, i.e., different human evaluators could give different evaluations on the same dialogues, which may lead to the unfairness in comparison among algorithms. This is another reason why the automatic metrics are necessary, since these are at least objective, so that it is fair to use these for evaluating different algorithms. However, the automatic metrics need to reflect the user experience more accurately and this should be urged in investigation and study in the future research.
>
> Finally, thank you for your comments again.
>
>
> Best Regards,
>
> Authors of paper 443
>
>
> References
>
> [1] Tiancheng  Zhao,  Kaige  Xie,  and  Maxine  Eskenazi.Rethinking  action  spaces  for  reinforce-ment  learning  in  end-to-end  dialog  agents  with  latent  variable  models.arXiv  preprintarXiv:1902.08858, 2019.

---

### Decision · Program_Chairs · 2021-01-07
**Final Decision**

**Decision:**

Accept (Poster)

**Comment:**

As the title states (and reads somewhat like an openreview review title), the authors apply the options framework from the RL community to perform hierarchical RL where the option is the dialogue act and the subproblem is the NLG component in task-oriented dialogue (TOD) policy learning. The two technical contributions (beyond the conceptual connection above) is showing that asynchronous updates between the hierarchy levels guarantees convergence and language-model based discriminator to densify the reward structure. Empirical results are solid improvements over recent SoTA findings.

== Pros ==
+ This is a conceptually appealing application of RL to TOD and they authors had to make additional modifications to get it to work — which will help other researchers in this space.
+ There are both theoretical and empirical contributions. The theoretical contributions are also insightful and not superfluous to the problem being studied.
+ Using a language-model based discriminator for reward shaping isn’t completely new (although I haven’t seen in this setting and stated exactly the same), but is interesting and effective.

== Cons ==
+ The writing could use significant work; while the reviewers/rebuttal cleared up many issues, I actually didn’t appreciate the value of this paper in my first read due to the writing (even if the motivation, etc. is sufficiently clear).
+ Human evaluation is treated as somewhat of an afterthought and there isn’t a deep dive into error analysis of the results. The visualization is a good first step, but there isn’t really a when/why this method works better than others, which is important for a problem where evaluation isn’t conclusive in the best cases. This is also significant since the authors claim ‘comprehensibility’.

Evaluating along the requested dimensions:

- Quality: The conceptual and theoretical contributions are both of high-quality. This is a promising approach to TOD and the authors additions (e.g., async optimization, LM reward shaping) are good examples of applied research. The empirical results are sufficient to above average, but not as strong (although this is partially an artifact of TOD evaluation)
- Clarity: The motivation is good, but the paper could use some work in writing. Some examples include (1) stating precisely how the option choices are derived (latent variables), (2) mapping out notation in something like a preliminaries section, (3) sketch of proofs in the main body for continuity. If the reader is familiar with the closest cited work, it is a bit easier, but I think some effort in making the paper more self-contained would increase its impact.
- Originality: Options in HRL is widely known, but applying it to TOD is novel to the best of my (and the reviewers) knowledge. I think many could have come up with the basic idea, but it took some effort to get it to work.
- Significance: This is a widely studied problem and the approach is fairly convincing. I don’t think it will be ‘disruptive’ or cross-pollinate to other application areas, but will almost certainly be cited within the conversational agent community.

In summary, the reviewers like this paper a bit more than myself personally — I think it is borderline with a preference to accept while the reviews are a more confident accept. However, the reviewers are also experienced experts in this area. I also do think that the authors handled concerns well in the rebuttal stages and addressed my more pressing concerns. I would encourage the authors to improve the writing if accepted, but I would prefer to accept this if possible.